# Is Wood a Material? Taking the Size Effect Seriously

**DOI:** 10.3390/ma15155403

**Published:** 2022-08-05

**Authors:** Stephen M. Walley, Samuel J. Rogers

**Affiliations:** Cavendish Laboratory, J.J. Thomson Avenue, Cambridge CB3 0HE, UK

**Keywords:** wood, size effect, Weibull, weakest link, strength, timber, lumber

## Abstract

This review critically examines the various ways in which the mechanical properties of wood have been understood. Despite the immense global importance of wood in construction, most understanding of its elastic and inelastic properties is based on models developed for other materials. Such models neglect wood’s cellular and fibrous nature. This review thus questions how well models that were originally developed for homogeneous and effectively continuous materials can describe wood’s mechanical properties. For example, the elastic moduli of wood have been found by many authors to depend on the size of the test specimen. Such observations are incompatible with classical elasticity theory. There is also much uncertainty about how well elastic moduli can be defined for wood. An analysis of different models for size effects of various inelastic properties of wood shows that these models only approximate the observed behaviour, and do not predict or explain the scatter in the results. A more complete understanding of wood’s mechanical properties must take account of it being in some sense intermediate between a material and a structure.

## 1. Introduction

To the ancient Romans (and probably many modern readers of this article!), the first part of the title of this article would be absurd. The Latin word *materia* from which the English word ‘material’ is derived meant timber [1]. Ashby also included wood as a material in his property maps [2].

Along with concrete and steel, one of the main uses of wood is in construction [3,4,5,6,7]. However, if you are going to design and build a structure, you require a good knowledge of the mechanical response of the substances you use [8]. However, testing machines large enough to measure the properties of the structural components of buildings are uncommon and expensive. Therefore, mechanical measurements are usually performed on small specimens. The assumption is then made that it is valid to extrapolate from data obtained in this way to the response of large objects (such as beams, columns and floors) to the loads they will be exposed to [9]. This methodology has long been found to be valid for substances such as metals [9,10]. This is because the granularity of metals is small enough that it can be ignored on the scale of their intended use. Wood, however, has both a tubular and a fibrous structure [11,12,13] (Figure 1). Timber can also contain locked-in strains, particularly if derived from the branches of trees [14,15,16]. The question then is whether the mechanical properties of wood can be measured using techniques that were developed for more homogeneous materials (such as metals), or whether wood’s mesoscale heterogeneity [17,18] means that it needs to be thought about in some other way.

The most thorough attempts so far to mathematically analyze the anisotropic elastic properties of wood taking into account its known structure were published by Price in 1929 [19] and by Kahle and Woodhouse in 1994 [20]. However, the equations both sets of authors derived are so complicated that as far as we are aware, they have been more often admired at a distance than actually used. The problem is that in order to use their results it is necessary to know the mechanical properties of the cell wall material in situ [20,21,22,23,24] because the indentation hardness of the cellulose/lignin combination that the cell walls of wood are made from depends on wall thickness [25]. The only plausible way of making such measurements is nanoindentation [26,27,28], a technique that was developed in the 1980s [29] and first applied to wood in 1997 [30,31,32].

In order to investigate the issues raised above, we start with a brief summary of the origins and underlying assumptions of classical elasticity theory. We then discuss the ways in which this theory has been applied to wood. This is followed by an overview of some key studies that show that both the elastic and inelastic mechanical properties of wood have been found to depend on the size of the specimen tested. We then present some size-effect models that have been developed for homogeneous materials and discuss their shortcomings for describing wood. We conclude by briefly considering whether insights obtained from the mechanical properties of cellular materials may provide a useful framework in which to consider wood. We also suggest topics that need further investigation.

## 2. The Classical Understanding of Elasticity

The concept of linear elasticity is usually credited to Robert Hooke, who, in the 1670s, proposed that for springs there exists a linear relation between the force applied and the extension produced [33]. About 130 years later, Thomas Young popularized the concept of linear elasticity in Lecture 13 of a lecture course he published in 1807 [34]. This resulted in the elasticity modulus, *E*, being named after him, which is defined as follows: (1)E=σ/ε
where σ is the true stress and ε is the true strain.

However, as Truesdell pointed out in 1960 [35,36] and Bell in 1973 [37], significant progress was made by other men in the understanding of elasticity during the 18th century. For example, Jacob Bernoulli realized in 1705 that “a *stress–strain relation* $\Delta l/l=f\left(F/A\right)$ characterizes a *material* rather than a particular specimen” [36] and Leonard Euler defined the elastic modulus in 1727 [38,39,40]. Nevertheless, Young’s thirteenth lecture provides helpful insights into the ways in which elasticity was understood in the early 19th century. 

Young assumed that materials are isotropic, meaning that the constant of proportionality between stress and strain is the same in all directions. He also focused his analysis on substances which have a texture that is “uniform, and not fibrous” [34] (p. 145). By ‘fibrous’, Young meant both what we mean by the term (he gave moist or green wood as an example of a fibrous substance [34] (p. 147)) and also, following Galileo [41,42,43], as a model for analysing the bending and torsional deformation of homogeneous substances such as metals. Young wrote as follows on pages 140–141 of his thirteenth lecture [34]: 


*“We might consider a wire as composed of a great number of minute threads, extending through its length, and closely connected together; if we twisted such a wire, the external threads would be extended, and in order to preserve the equilibrium, the internal ones would be contracted…”*


During the 1820s to 1840s, Eaton Hodgkinson reported the results of some studies that he and other men had performed on the strengths of cast iron, steel, stone and timber in connection with a number of construction projects, particularly bridges [44,45,46]. A summary table of some of his findings (which he quoted to an unjustifiable level of accuracy) is presented as Table 1. Most of these measurements were made on site rather than in testing workshops, which were usually a long way from where the information was immediately needed [47]. Note that Hodgkinson did not report the information in terms of stress, but following Girard [42] (Figure 2) as the load at which a pillar (i.e., rod) of a given material broke (see, for example, Figure 3). 

It should be noted that from at least the early 1860s, rocks and metals were known to contain grains [49,50], although microstructural images demonstrating their granularity do not seem to have been published until the 1880s (Figure 4).

According to Bell [10,54], an important set of careful tests that established linear elasticity for “small quasistatic deformation” of metals were those performed by Alphonse Duleau in the 1810s [10,55]. However, around the same time Pierre Dupin showed that the deflection, δ, of the centrally loaded wooden beams had a quadratic rather than a linear dependence on load, *F*, [56,57]: (2)$$\delta =bF+cF^{2},$$
where *b* and *c* are constants for a given type of wood. It should be mentioned here that since a major use of wood in service is as beams, the study of wood in bending or flexure (which is a mixture of the ‘pure’ states of compression, tension and shear) is important and will be considered in more detail later. 

Both men were motivated by practical problems: Duleau had been commissioned to design a bridge out of iron and Dupin had been tasked to investigate the deformation of wooden ships [58,59]. 

Extensive further experimentation during the 19th century helped to establish the laws of linear elasticity for many materials [37,60,61,62,63,64], so that by the early 20th century, the theory of linear elasticity had gained widespread acceptance as well as considerable mathematical sophistication [65,66,67]. This was despite it being well known that the true elastic (that is to say, recoverable) response of a wide variety of substances is nonlinear [10,68].

The basic question then that underlies this review is what needs to be true of a substance for elasticity theory to be applicable, even if only approximately. In 2013, Christensen discussed this issue for failure [69]. He wrote as follows: 

“Well-constructed failure theories can discriminate safe states of stress in materials from states of certain failure, based upon calibration by a minimal number of failure-type mechanical properties. The specific purpose here is to provide failure criteria for general types of materials. Two of the conditions that are taken to apply are those of a macroscopic scale of consideration and the corresponding macroscopic homogeneity of the material.

The concepts of macroscopic scale and macroscopic homogeneity have connotations familiar to everyone. However, trying to define these concepts in absolute terms is extremely difficult. Macroscopic homogeneity is taken to be the condition that the material’s constitution is the same at all locations. Thus, the problem is shifted to the precise meaning of the term ‘location’, which depends upon the scale of observation. Suffice to say, the scale of observation is taken such that all the common forms of materials are included, such as metals, polymers, ceramics, glasses, and some geological materials. Materials which are excluded are porous materials, whether cellular or not, as well as granular materials.”

Christensen thus implicitly excluded wood from his theory of failure, since wood is both porous and cellular.

In this review, we will take the term ‘material’ to mean a substance that has mechanical properties that are independent of the size of the object made from it [9].

## 3. Problems with the Application of Elasticity Theory to Wood

Historical reports of mechanical tests on wood assume that classical elasticity theory can be applied [19,70,71,72]. Thus, student textbooks about wood written from the 1950s to the 1990s assumed that its mechanical properties can be measured, reported and used similarly to any other material. 

For example, in 1996 in the seventh edition of Desch and Dinwoodie’s student textbook on timber [73], they introduce the mechanical properties of wood by defining a limit of proportionality below which the deformation is linearly proportional to the applied stress, the constant of proportionality being the Young’s modulus (Figure 5a). Beyond this limit, they state that subsequent deformation is not recovered upon removal of the stress (Figure 5b). The implicit assumption they were making was that wood mechanically behaves similar to a ductile metal [74] and can thus be described as elastic–plastic. Many people who study the mechanical properties of wood are still making the same assumption up to the present day [75,76,77,78]. However, as Figure 6 and Figure 7 show, the compressive stress–strain response of wood is similar to that of polymer foams, albeit stronger. Thus, after a certain stress is reached, both wood and polymer foams deform at almost constant stress until their cellular structure has been destroyed (‘fully dense’), after which the stress climbs rapidly as deformation proceeds.

Note also that due to the way trees grow, its mechanical properties depend on the angle between the loading direction and the grain [4,81,82,83,84] (Figure 7, Figure 8 and Figure 9).

Wood’s response to loading also depends on whether it is loaded in compression or tension (Figure 10 and Figure 11). Additionally, generally speaking, ‘straight-grained wood’ is stronger in tension than compression [86]. Figure 10b and Figure 11 also show that even clear wood usually fails in a brittle manner under tension. The difference between the mechanical response in compression and tension produced by bending is even more pronounced for lumber/timber because of the presence of knots (Figure 12 and Figure 13) [87].

One major problem with testing wood in tension is gripping the specimens, which are usually of the standard dog-bone design (Figure 14) as developed for metals [88]. For as Dinwoodie pointed out: “[Tension tests are] performed only infrequently as the amount of timber loaded in tension under service conditions is quite small. A further reason for the lack of tensile data is the difficulties experienced in performing the tensile test: first, due to the very high tensile strength of timber, it is difficult to grip the material without crushing the grain, especially in low-density timbers; and second, in timber with very high tensile strength, failure is frequently in shear at the end of the waisted region rather than in tension within the waisted region. It is very difficult to conduct the standard tensile test in green timber” [89].

One striking comment that Desch and Dinwoodie made in their textbook is that the Young’s modulus is “a material constant characterizing one piece of wood”. If true, this statement makes the concept of Young’s modulus for wood almost useless, for elasticity theory is of little use in designing large structures out of a substance if its moduli are only known for individual specimens. In practice, this problem is hidden from sight because engineers and architects design structures with large margins of safety in order to cope with the imprecision with which mechanical properties are known for building materials, let alone how their properties change during their service lifetime [90,91,92,93,94,95,96,97,98,99].

Desch and Dinwoodie go on to say that the modulus “will be similar for other samples from the same part of the tree”. Such observations naturally lead on to the idea that the strength of wooden beams varies along their length, largely due to imperfections (or growth ‘defects’) such as knots (Figure 15, Figure 16 and Figure 17) which are formed where branches connect with other branches or the trunk of the tree [86,100,101]. Madsen also reported an effect of beam length on strength even for knot-free (i.e., clear) wood [86,102,103,104]. For these reasons, a distinction is usually made between ‘wood’ (termed defect-free or ‘clear’) and ‘lumber’ or ‘timber’, which contain knots. Madsen showed that the size effect is more pronounced for timber the more knots it contains [102]. The effect on the tensile strength of a wooden beam of the number of weak sections it contains was subsequently analysed and quantified by Kohler and co-workers (Figure 18) [101].

Figure 15 and Figure 16 also show schematically that one consequence of the cross-sectional strength varying along the length of a beam is that failure rarely takes place where the applied stresses are a maximum but can occur anywhere along the length [102]. This is because the probability is very low that a critical defect (or ‘weakest link’) will occur where the stress is at a maximum. One major implication of this is that the length of a beam will have an effect on the strength that is measured. Madsen also found that wooden beams are stronger in bending than in compression or tension [102]. This is because when a beam is bent, only about 10% of its volume is subjected to high stresses, whereas when a beam is loaded in pure compression or tension, all of it is subjected to the same stress.

## 4. Problems in Discerning Trends in the Strength of Wood

Madsen and Buchanan pointed out that one major problem with checking theories of the mechanical properties of wood is that measurements of the strength of timber have a large scatter [86] (see, for example, Figure 19, Figure 20 and Figure 21). One major cause of this variation is that cutting a piece of timber into small pieces for mechanical testing (e.g., Figure 14) will produce some specimens that are defect-free (clear), whereas others will contain knots (Figure 22). Other sources of intrinsic variation in wood’s mechanical properties are moisture content and specific gravity (Figure 23), the species from which the wood was obtained [105,106] and the circumstances under which the tree grew [107,108,109,110,111]. Extrinsic factors such as the duration of loading can also have a large effect (Figure 24). It is notable that compared with the metals or ceramics literature, information about specimen preparation (such as surface finish) is usually lacking from the wood literature, although when photographs of specimens are provided (e.g., Figure 14), care does seem to have been taken in their preparation.

We have already mentioned in the discussion of Figure 15, Figure 16 and Figure 18 that long wooden beams will likely be weaker than short ones. The data presented in Figure 22 hint that this ‘size effect’ can be produced by any dimension of a piece of timber, not just its length.

## 5. Size Effect Theories

The variability of measured mechanical properties with specimen size is neglected in the classical theory of strength which assumes that “the mean strength obtained from a number of geometrically similar tests [is] the measure of material strength” [113]. 

Mathematically, size effects are often described by the Weibull distribution [114], which is one example of a weakest-link theory [115,116]. The big idea of weakest-link models is that the overall strength of an object (such as a beam) depends only on the strength of its weakest section, for once the stress in a section reaches the value needed to break that section, the entire object will end up broken [9]. Williams pointed out back in 1957 that artisans have known for centuries that (all other things being equal) short ropes are stronger than long ones even though luminaries such as Galileo and Young convinced themselves by the application of logic that this cannot be so [117,118,119]. 

As the name suggests, Weibull weakest-link theory was first developed by Weibull, his first papers on this topic being published in 1939 [120,121]. He then reassessed the theory in the early 1950s [122,123]. His analysis showed how the strength of a system is described by a cumulative exponential distribution; therefore, the strength depends on the specimen dimensions in the following manner: (3)$$\sigma_{N}\propto D^{-\left(n_{d}/m\right)}$$
where $\sigma_{N}$ is the nominal strength, *D* is the size of the specimen (usually its length), $n_{d}$ is the number of dimensions in which the structure is scaled (usually 2 or 3), and *m* is an experimentally determined parameter [9,124,125]. A simple derivation of the above formula was given by Barrett in 1974 [113]. 

It should be noted here that size effects exist for all substances, since the larger the object, the greater is the probability that it will contain a critical flaw. As a result, size effects are being actively investigated for a wide range of material types [9,96,124,125,126,127,128,129,130,131,132,133,134,135,136,137,138]. Therefore, Weibull’s analysis can be applied to a beam made of steel with a set of defects randomly scattered along its length. Thus, in 1994, in a paper about the application of Weibull’s theory to the problem of the strength of materials, Lindquist wrote [139]: 


*“Predicting yield of structural members under complex loading conditions is a difficult task for the engineer. Complex loading often results in the structural members being stressed biaxially or even triaxially, whereas yield strength data are usually only available for tests conducted in uniaxial (tensile or compressive) or torsional stress states. The test specimens are also typically much smaller than the actual structural members. The problem, therefore, is to predict structural member yield using only these uniaxial and/or torsional yield test results. The problem of relating the test results in simple stress states to full-scale members under much more complicated stress conditions is often solved using what is known as the maximum distortion energy theory.”*


In this theory, the uniaxial distortion energy *u* is given by:(4)$$u=\sigma^{2}/6G$$
and the torsional distortion energy *u* is given by: (5)$$u=\tau^{2}/2G,$$
where σ is the uniaxial stress, τ is the torsional stress, and *G* is the shear modulus.

Lindquist then discussed the classic ideas of Huber, Hencky and von Mises, who analyzed elastic energy as being the sum of two parts: dilatation and distortion. Their basic idea is that when the distortion energy reaches a critical value, the material yields. This concept validates the use of data obtained in uniaxial and torsion stress tests when the state of stress is complicated. 

Lindquist then introduced Bayesian probability as follows:


*“One must now consider how the test results on small samples of material relate to the full-scale structural members. In many cases the yielded volume in a failed full-scale member will be orders of magnitude larger than the yielded volume in the average test specimen. It therefore seems reasonable that each test specimen’s distortion energy capacity can be considered a point measurement of the distortion energy capacity for large members. An engineer would therefore be interested in using the distribution of the mean distortion energy capacity of the material $\left(\theta \right)$ rather than the distribution of the test sample distortion energy capacities as a design guideline.”*


Additionally, there are two other aspects that need to be properly defined for each case if prediction is to be achieved: the characteristic energy and the characteristic length-scale (which defines the volume). The fact that Lindquist chose the von Mises distortional energy shows that he was being guided by metals thinking.

Later, Porter and co-workers essentially followed Griffith [140,141] and used the same probability argument but without using Bayesian analysis [142]. This means that the energy can be characteristic of any of the dissipation mechanisms open to a material and the length-scale is thereby chosen from the dissipation mechanism. 

The important thing about their argument is that a material can have many modes of energy dissipation which all act at the same time. Dissipation is triggered by the activation of the most probable. This may then add or remove dissipation mechanisms. Therefore, for example, a brittle material starts with a single mechanism (crack initiation) but can then develop a second mechanism (crack growth). Porter et al. considered that all of these dissipation mechanisms were controlled by the point of inflection of the relevant volumetric potential function. 

About the same time, Christensen proposed a single criterion for all failure-like mechanisms that could have made use of this idea [143]. Christensen’s idea relies on a definition of failure which is atomistically local and is defined by the conversion of stored elastic energy to some other form usually associated with irreversible deformation. Note that Christensen explicitly said that his theory/methodology of failure did not cover “cellular materials (foams), granular materials, and other inhomogeneous materials forms. Their failure criteria require separate development” [69,144].

A number of reasons for the existence of size effects in mechanical testing have been identified: (i) friction in both quasistatic [145] and dynamic compression testing [146] (testing in tension or bending can move around this problem); (ii) inertia in dynamic testing, whether in tension or compression [146]; (iii) the distribution of flaws [120]; and (iv) an internal cellular or fibrous structure [4].

## 6. Evidence of Size Effects in Wood

Weibull considered many materials, but wood was not among them. It was not until the 1960s that Weibull’s weakest link analysis was first applied to wood by Bohannan [147,148] who was also the first to report a size effect in the mechanical testing of wood [100]. Before then the strength properties of timber/lumber had been derived from small-scale tests on clear wood, which were then corrected for variables such as moisture content, load duration, etc. However, this way of doing things did not take into account the fact that clear wood (by definition) contains no visible defects whereas timber/lumber contains growth defects such as knots resulting in differences in failure mode between wood and timber. Therefore, in the 1980s, full-scale testing of representative specimens was recommended in which the load was determined at which 10–15% of the samples would break. It was discovered that deep-bending and wide-tension members are weaker than smaller counterparts, confirming that there is indeed a size effect for wood. As there is visible variation in the mechanical properties along the length of a wooden beam, it made sense to turn to Weibull’s analysis (a well-established weakest link theory) to describe it [100,128].

Barrett assumed (for simplicity) that for wood “…all variability in load-carrying is due to natural material variability” and “A complete evaluation of risk of failure would necessarily require a thorough knowledge of statistical variation of load quantities” [113]. Barrett also reported that as far back as 1956, Markwardt and Youngquist had observed differences in the strength of differently sized specimens of wood loaded in tension but offered no explanation [149]: “Their results presented show that strengths obtained are specimen-dependent, which makes evaluation of material properties extremely difficult.” For example, they found that halving the width of a specimen prepared according to ASTM standards increased the strength of Douglas Fir from 254 to 312 psi (1.75 to 2.15 Mpa) (Figure 25).

Over the next 20 years or so, Barrett and co-workers studied bending, tension and compression properties of Canadian softwoods parallel to their grain in order to quantify the size effect for each of these three modes of loading [154]. They found there was a slight tendency for the size factor to decrease with increasing modulus of rupture (Figure 26). They also reported that for visually graded lumber, size effects were equal (at the 5% significance level) across grades and tree species. Length effect factors for tension and bending were similar. Width effect factors for bending tests were slightly higher than for tension factors.

As mentioned earlier, the central objection to the classification of wood as a material in the classical sense is the evidence that the mechanical properties of wood vary with the size of objects made from it (Figure 22, Figure 25, Figure 27 and Figure 28). 

A great deal of research has been and is being performed on the effect of specimen size on the proportional limit and fracture stresses of wood. For example, Dinwoodie stated on page 194 of his book about timber [89] that “Timber appears to exhibit size effects to a greater extent than most other materials” and that Barrett had shown there is a relation “between specimen volume and strength for timber loaded in tension perpendicular to the grain” [113]. However, in that paper (published in 1974), Barratt also stated that Weibull’s distribution had “does not appear to have been widely applied in studying wood mechanical behavior”.

Analysis of the bending strength of *Eucalyptus grandis* samples showed that their strength decreased with increasing specimen depth, despite there being significant variation in the measurements obtained from specimens of the same size (see Figure 29) [130].

Tests performed by Zauner and Niemz using cylindrically symmetric specimens of Norway Spruce (*Picea abies*) (Figure 30) also demonstrated a clear decrease in strength with increasing specimen size (Figure 31) [125]. The three theories they investigated were (i) Weibull’s Weakest Link Theory (WLT) (Equation (3)), (ii) Bazant’s Size Effect Law (SEL) which is based on linear elastic fracture mechanics (Equation (6)) [156], and (iii) Carpinteri’s Multi-Fractal Scaling Law (MFSL) which is based on geometric arguments (Equation (7)) [157].
(6)$$\sigma_{N}=\frac{B}{\sqrt{1-\frac{D}{D_{0}}}}$$
where *B* and *D*_0_ are experimentally determined constants.
(7)σN=A+BD,
where *A* and *B* are experimentally determined constants.

As the various numerical factors in these three equations are determined from the experiments they performed, it is not surprising that all three equations fitted the data they obtained. The main difference is that the Weibull plot is a straight line through the highly scattered data, whereas the other two theories deviate in different directions for specimen sizes larger and smaller than those shown in Figure 31. Note that the data and fits are presented in this figure using a log–log plot. Hence, the scatter in the data is even worse than it appears in the plot. To conclude, is not possible to tell from these tests which of the three size effect laws they considered is the best, and one may as well go with the simplest fit, which is the Weakest Link (or Weibull) Theory.

Many other papers exist which corroborate these results, such as those shown in Figure 32. Therefore, although there has long been debate about which mathematical law best describes the size effect (Figure 33 and Figure 34) [138,158,159,160,161], there is widespread consensus that the strength of wood decreases as the specimen size increases (Figure 34).

MFSL theories are based on a model of brittle failure due to the propagation of microcracks. The idea is that below some critical strain microcracks do not propagate sufficiently to have any effect at the macro scale, but the correlation length of these cracks grows to infinity at the critical strain [157]. Carpinteri and Chiaia treated critical failure using the framework of phase transitions, arguing that at the critical point the system has similar fluctuations on all length scales and therefore no characteristic length can be associated with this process. Such self-similarity properties make the system analogous to a mathematical fractal. This analysis results in an expression which scales the nominal tensile strength, $\sigma_{N}$, with one of the dimensions, *D*, of the specimen (see Equation (7)).

Bazant’s Size Effect Law, in contrast, is based on the observation that a well-defined Fracture Process Zone (FPZ) exists for all ‘quasi-brittle’ materials; hence, there is a characteristic size scale associated with fracture [156]. This model results in the relation presented in Equation (6), which relates the fracture strength, *σ*_N_, to the size of the specimen, *D*, scaled in terms of a parameter *D*_0_.

The Size Effect Law was first tested against data for the fracture stress of concrete [163], but has since been tested for wood [162,164]. While there is still no complete consensus, the Size Effect Law seems the best candidate for a general law to describe the scaling of fracture properties.

However, since the 1960s, Weibull’s theory has been widely used for wood, albeit with some modifications, and has had some success in modelling the observed size effect data [130], although Bazant and Yavari have pointed out a number of problems with it [9].

As mentioned before, the mechanical properties of lumber are dominated by a small number of large defects whereas those of (clear) wood are governed by a large number of small defects [103]. Additionally, in lumber, the distribution of defects along the length may be different to those in the other two dimensions, i.e., depth and width. Therefore, a two-parameter Weibull distribution is needed with the boundary condition of zero strength for infinite size.

In 1986, Madsen and Buchanan analysed the length effect using brittle fracture theory [86]:(8)$$\frac{x_{1}}{x_{2}}={\left(\frac{L_{2}}{L_{1}}\right)}^{1/k_{1}},$$
where $x_{1}$ and $x_{2}$ are the strengths of beams of length $L_{1}$ and $L_{2}$. This function is plotted in Figure 35 for $\frac{L_{2}}{L_{1}}=0.5$. The experimental data that they obtained for Canadian Spruce are shown in Figure 36. This figure also shows that they found that strength *increased* with depth, contrary to their theory. At that time, they had no explanation for this observation.

Madsen and Buchanan also took a new approach to the study of size effects by considering bending [86]. Bending is the mode of loading that most wood used in construction is subjected to (up to that point, only shear and tension had been included in Canadian design standards). Figure 37 shows the differences between the tensile, compressive and bending strengths. The bending strength can be seen to be intermediate between tensile and compressive strength, which makes sense since bending is a mixture of tension and compression. Figure 38 shows that wood is brittle in tension and ductile in compression (see also Figure 10 and Figure 11). Failure in bending can be either brittle (Figure 39) or ductile (Figure 40).

The most thorough study performed so far of size effects in timber was carried out by Madsen and Tomoi [104]. They studied wood from three different species of tree (spruce, pine and fir), each cut into 27 different length, breadth and depth combinations (Figure 41). They tested at least 100 specimens for each test configuration. Their testing programme used clear (i.e., knot-free) wood.

If Weibull’s theory is true, log(strength) will be linearly related to log(length) [86]. They found that beam length was of primary importance (Figure 42). There was a weak (or inconsistent) effect of depth (Figure 43).

In the 1960s, Bohannan found that for defect-free material, data were fitted best by graphs of log(strength) vs. log(aspect ratio), where aspect ratio = (length times depth) rather than log(volume) as is suggested by Weibull’s theory [147]. However, although Madsen found that the strength of shorter wooden beams is greater than that of longer ones [103], he found that his experimental data were fitted better by log(strength) vs. log(volume) (Figure 44) [103]. Figure 44 also shows that the size effect was smaller for wet wood as opposed to dry wood, and that “a length effect could not be found for wet material”. Even for dry tests, there was only a 5% reduction in strength for when the length was doubled. The mode of loading was also found to be important, the length effect being small for compression compared with tension and bending. Madsen found the way the load was distributed along the beams (Figure 45) was very important, but he did not quantify this effect.

Most size effect studies have been performed quasistatically. The results of one very recent study of the impact fracture of two different woods (Figure 46) shows that the size effect may be more complicated in impact than at low rates of strain (Figure 47) [166].

The weakest link theories discussed so far have assumed that flaws are uniformly distributed, but due to the way trees grow, flaws are arranged anisotropically in timber/lumber [86]. Thus, different distribution functions will be needed for different directions. As an example of this, Madsen observed an effect of length but not of depth on strength. Quantifying this, he found an 18% reduction in strength when the beam length was doubled. To summarize, Madsen found that (i) the size effect can be described by a parameter *g* equal to the slope of the graph of log(strength) and log(size) (Figure 48); (ii) *g* was 0.22 for tension, 0.10 for compression and 0.20 for bending; (iii) the length effect did not depend on depth for the widths he tested.

The scaling of the elastic properties of wood with specimen size has been studied by only a few people, but those papers that do exist on this topic demonstrate that the effect is real, although the evidence is contradictory as to what the trends are [134,167]. For example, measurements made by Hu et al. on *Fagus sylvatica* showed that increasing the height of the specimen increased the modulus, whereas increasing the cross-sectional area decreased the modulus [134] (Table 2 and Figure 49).

However, a similar study by Xavier et al. on *Pinus pinaster* showed the opposite effect: increasing the specimen height decreased the modulus and increasing the cross-sectional area increased the modulus [167], as shown in Table 3 and Figure 50. They showed conclusively that this was due to friction between the specimen ends and the anvils used to compress the wooden cylinders.

The problem of data scatter is well demonstrated in plots of various strength parameters obtained using standard-sized specimens against the same parameters obtained using micro-sized specimens (Figure 51) in tension, compression and bending. Figure 51 also shows that the size effect is more pronounced for tension as compared to compression. Micro-sized specimens are increasingly being used to minimize the amount of wood taken from a structure for testing [168]. However, as we have been at pains to point out in this review, there are serious concerns with using small specimens (it will only work if the mechanical properties at two different size scales are well-correlated).

Zhou et al. analyzed the effects of varying both lumber grade and specimen width on the elastic modulus of Chinese Larch (*Larix gmelinii*), [131]. Grading was performed visually on the basis of the observed defects. Their data do not show any compelling overall trends, and there is a large variation found within the data obtained from a single grade and width (Figure 52) [169].

There are few studies which report measurements made for compression, tension and bending moduli for the same sample of wood. However, tests performed on man-made cellular substances show different size effects for compression, bending, shear and torsion of the same material, an observation which casts further doubt on the validity of the concept of elastic modulus to such substances [133].

The effect of size on wood’s mechanical properties can be reduced by gluing small pieces of wood together so as to create a more homogeneous product [170,171,172], but it is not always practical to do this.

## 7. Modelling Wood

The data for wood summarized in this review support the idea that an elastic modulus can only reliably be used to characterize an individual specimen, rather than a species of wood in general, due to the large variation observed from specimen to specimen. However, as discussed in Section 3 (Problems with the application of elasticity theory to wood), much of the literature on the strength properties of wood has assumed that its mechanical properties can be described using concepts developed for materials that are effectively homogeneous and continuous. Significant discrepancies with such theories, particularly the influence of the size of the specimen being tested, suggest that standard material models do not adequately describe wood. This section will therefore present three theories that have been developed to describe size effects in materials, and evaluate how well they apply to wood.

A number of studies have sought to model wood on several different scales, from nano to macro, in order to describe its response to mechanical loads [12,13,20,173,174,175,176]. For example, Zhan and co-workers suggested a representative volume element approach [13] (Figure 53), whereas Guindos and Guaita used geometrical approximations to the shapes of knots (three-dimensional growth defects) with some success (Figure 54 and Figure 55). The size of knots has also been found to have an effect on strength [177].

Another category in which wood is often discussed is that of cellular structures, the theory of which has been and is being developed for metal, polymer and ceramic foams; however, it should be noted at this point that wood has a tubular rather than a cellular morphology. This seems a promising method to describe those aspects of wood which quasi-brittle theories cannot, since whereas quasi-brittle theories were not developed for substances which have a mesoscale repeating substructure (such as wood), this is the central focus of cellular models [18].

One of the main insights to be taken from this body of literature is the strong dependence of bulk properties on boundary conditions in materials with heterogenous repeating structures. Wheel et al. used a simple model of a beam consisting of alternating layers of two materials with different moduli [179]. By simply altering the geometry of the setup, the model predicted opposing size effects, some geometries exhibiting an increase in strength with a decrease in size (‘stiffening effect’) and others a decrease in strength with a decrease in size (‘softening effect’). Wheel et al. concluded that “the circumstances determining the nature of the size effect appear to be governed entirely by the surface state of the material” [179].

Surface effects were found in other investigations. For example, Anderson and Lakes showed that open cells at the surface of a polymer material resulted in a softening effect [180]. Karakoç and Freund simulated experiments performed on the cellular structure of *Picea abies*, and concluded that the observed softening effect is the result of boundary effects, specifically the presence of stress-free walls at cell boundaries [132]. In 2018, in an overview of research into size effects in lattice structures, Yoder et al. argued that the non-homogenous nature of a cellular material, especially the difference in behaviour near a stress concentrator, means that attempts to model cellular substances as continuous materials are inherently flawed [133]. Against this, Tekoglu and Onck argued in 2005 that in the limiting case of a large number of cells, a foam can be approximated as continuous [129].

To summarize, this body of research into cellular materials may provide insights into the mechanical properties of wood, as long as its tubular and fibrous structure is taken into account.

## 8. Conclusions and Matters for Further Study

Significant size effects have been observed for the mechanical properties of wood. As a result, wood does not meet the criteria for being considered as a material in the sense that that the mechanical response of wooden structures cannot be predicted from performing mechanical tests on small specimens from the same source. Wood, therefore, should be thought of as being an intermediate between a material and a structure.

The main way of reducing the size effect for wood is cross-lamination, but it is not always practical or possible to do this.

While Bazant’s size effect law provides a good approximation to inelastic size effects in wood, a more accurate model would consider wood in a category of its own distinct from other quasi-brittle materials (such as concrete) and focus on its fibrous and tubular structure. The published literature on artificial cellular materials provides a promising body of research to gain insights from, bearing in mind that these differ from wood in mostly being three-dimensional foams.

There is also scope for further experimental investigation of the effects of different specimen geometries on the mechanical properties of wood.

## Figures and Tables

**Figure 1 materials-15-05403-f001:**
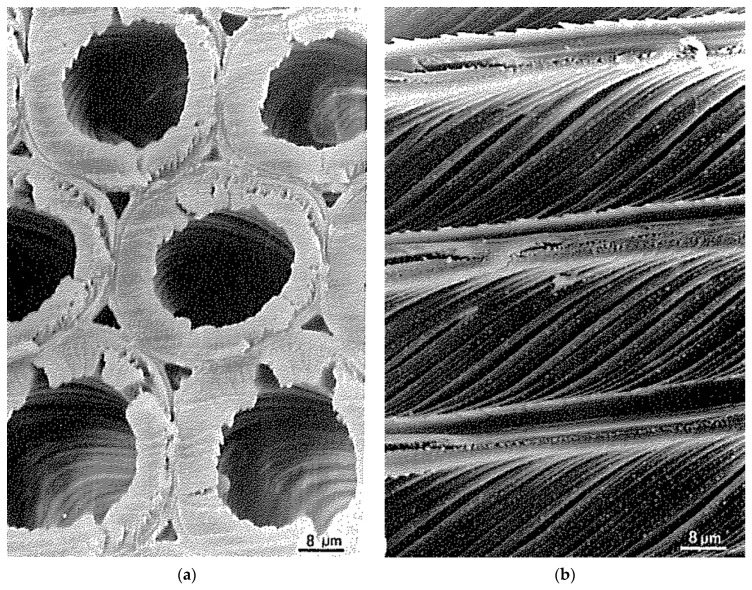
Electron micrographs of tracheids in the xylem of Radiata Pine (*Pinus radiata*). (**a**) Transverse section. (**b**) Longitudinal section. From [11].

**Figure 2 materials-15-05403-f002:**
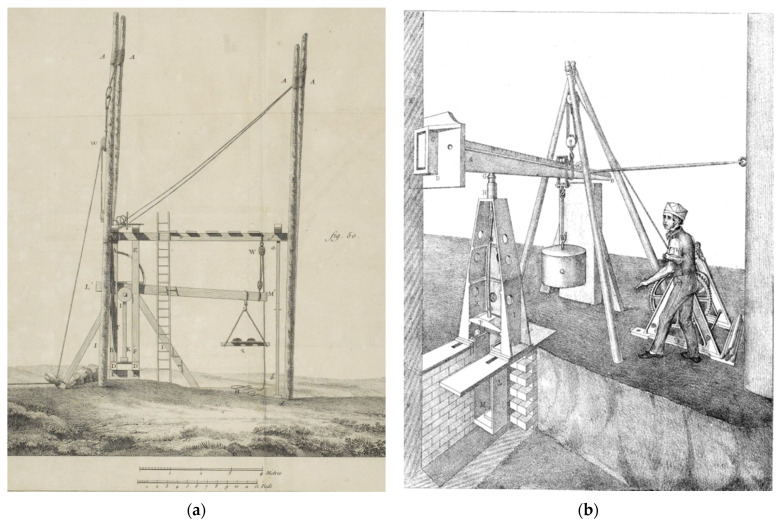
Examples of late 18th/early 19th century on-site testing machines for determining the compressive breaking loads of pillars (rods) of cast iron, steel and timber. (**a**) From [42]; (**b**) from [48].

**Figure 3 materials-15-05403-f003:**
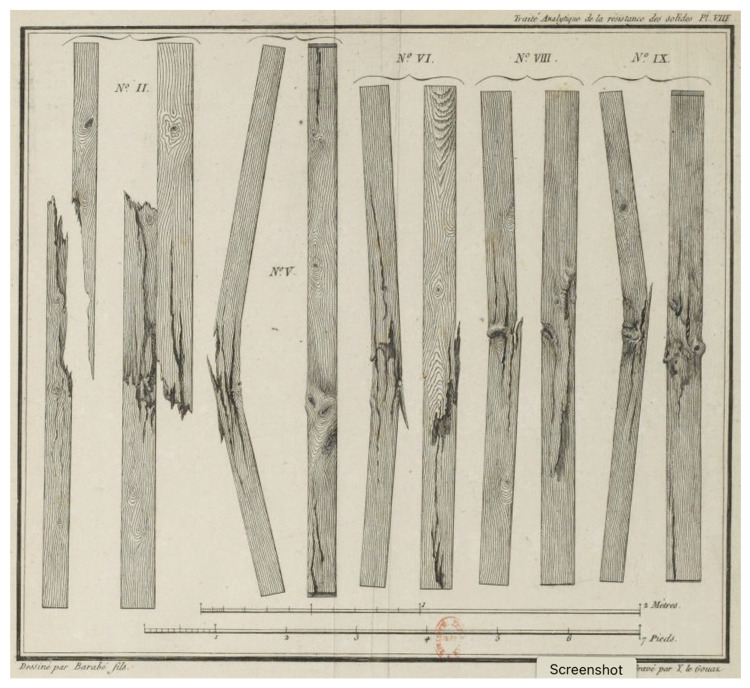
Engraving showing wooden beams broken by applying a longitudinal load using the apparatus shown in Figure 2a. From [42].

**Figure 4 materials-15-05403-f004:**
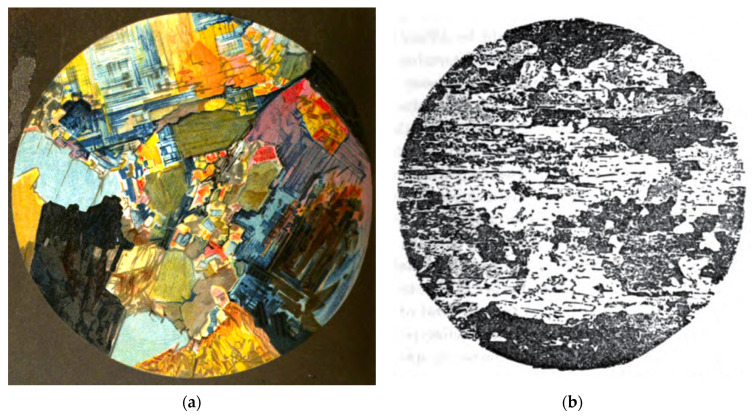
(**a**) Colour plate reproduction of a painting of the granular structure of granite observed in 1883 using polarized light microscopy. No scale bar or magnification was given. From [51]. (**b**) Optical micrograph of the microstructure of acid-etched armour steel published by Bayles in 1883 [52]. This photograph was originally taken by Sorby and presented at a lecture he gave to the Sheffield Literary and Philosophical Society [53]. No magnification or scale bar was included.

**Figure 5 materials-15-05403-f005:**
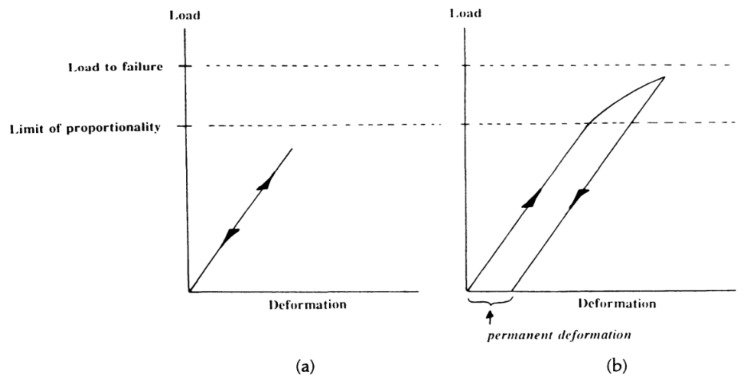
Schematic student textbook plots showing load-deformation paths for (**a**) the elastic (recoverable) and (**b**) the inelastic (irrecoverable) deformation of wood. From [79].

**Figure 6 materials-15-05403-f006:**
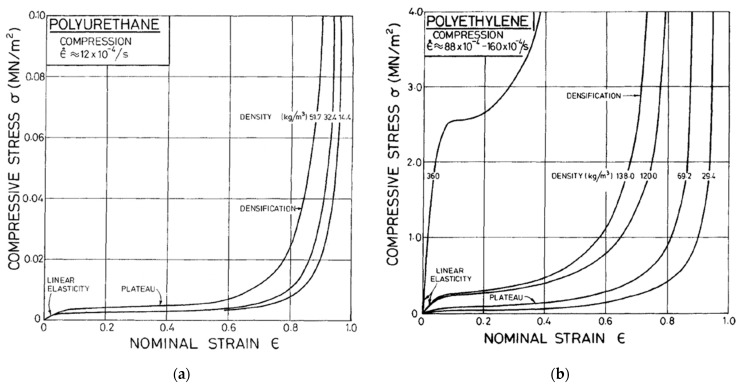
Experimental compressive stress–strain plots of (**a**) polyurethane foam tested at one strain rate and (**b**) polyethylene foam tested at several (low) strain rates. From [80].

**Figure 7 materials-15-05403-f007:**
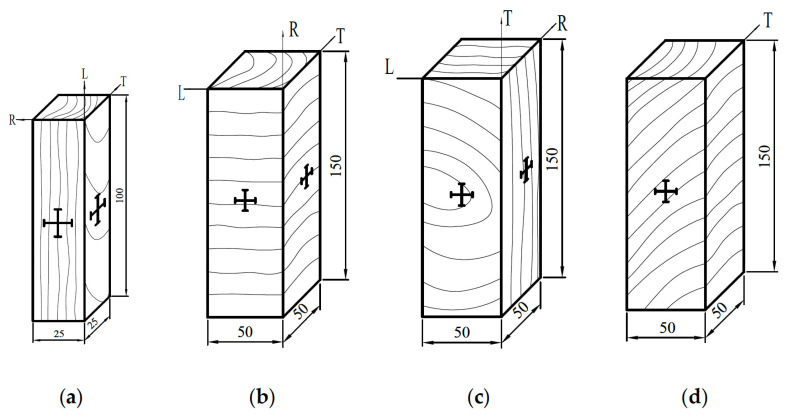
Schematic diagrams of rectangular wooden compression specimens cut at various angles to the grain of the wood. (**a**) Parallel to the grain. (**b**) Perpendicular to the grain radially. (**c**) Perpendicular to the grain tangentially. (**d**) at 45° to the grain. The ‘+’ signs indicate the position of strain gauges. From [84]. R means radial, L means length, T means tangential.

**Figure 8 materials-15-05403-f008:**
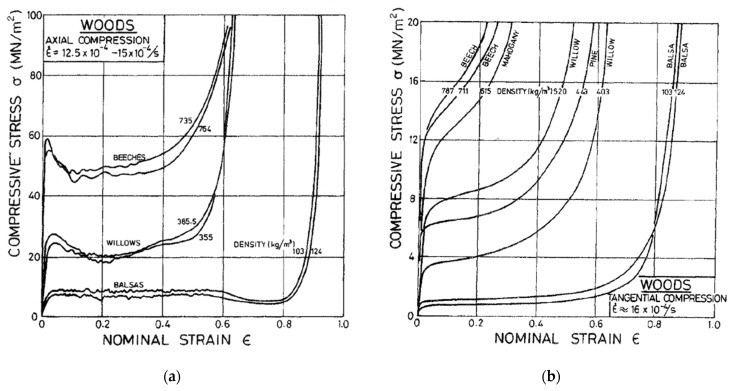
Experimental compressive stress–strain plots of a number of different woods measured using specimens (**a**) cut parallel to the axis of the trunk of the tree and (**b**) cut at a tangent to the trunk. From [4].

**Figure 9 materials-15-05403-f009:**
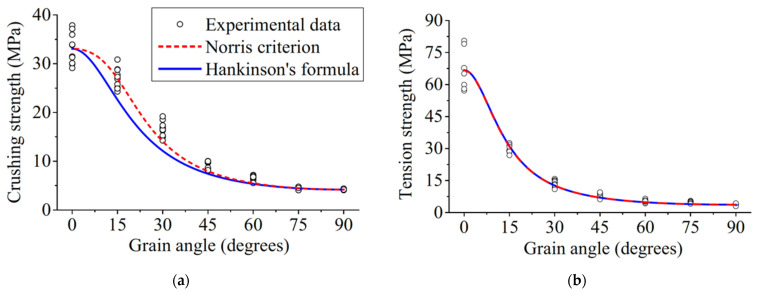
Strengths of Korean Pine as a function of grain angle in (**a**) compression and (**b**) tension. From [85].

**Figure 10 materials-15-05403-f010:**
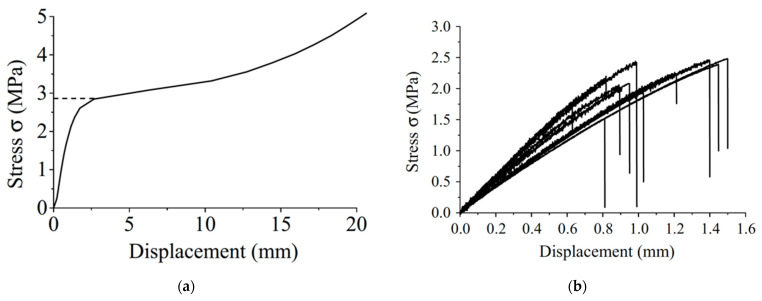
Stress-displacement plots for Korean Pine in (**a**) compression and (**b**) tension. From [85].

**Figure 11 materials-15-05403-f011:**
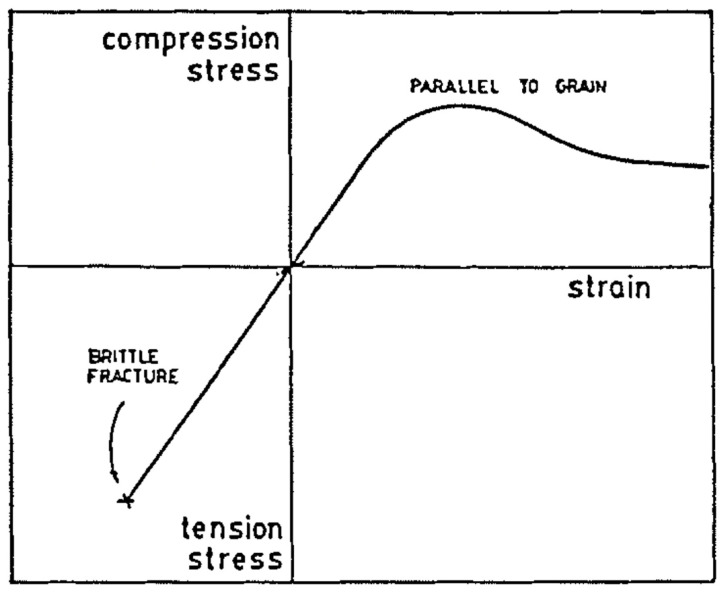
Schematic diagram showing the asymmetry of the tensile and compressive response of defect-free (or ‘clear’) wood. From [86].

**Figure 12 materials-15-05403-f012:**
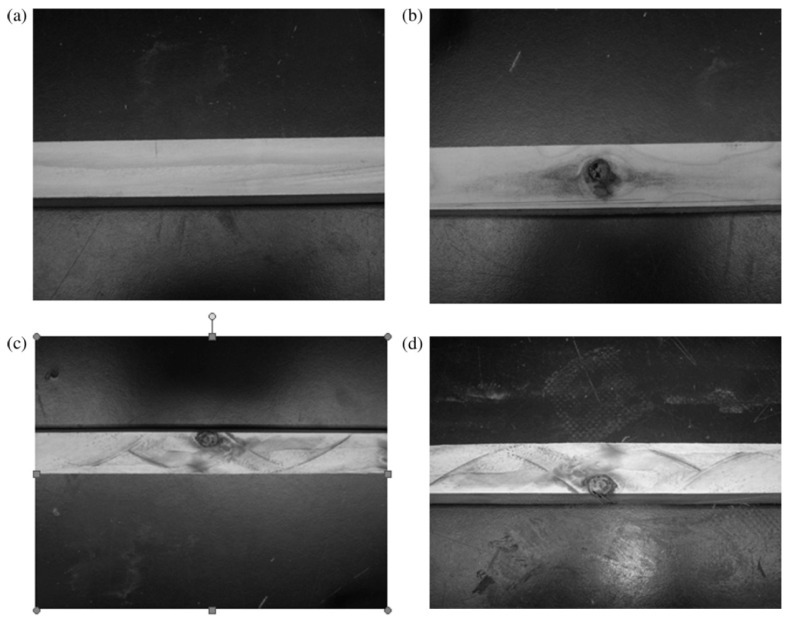
Photographs of four 25×25×206 mm Douglas Fir beam bending specimens. (**a**) Clear wood; (**b**) knot mid-height of the face; (**c**) knot subjected to compression; (**d**) knot subjected to tension. From [87].

**Figure 13 materials-15-05403-f013:**
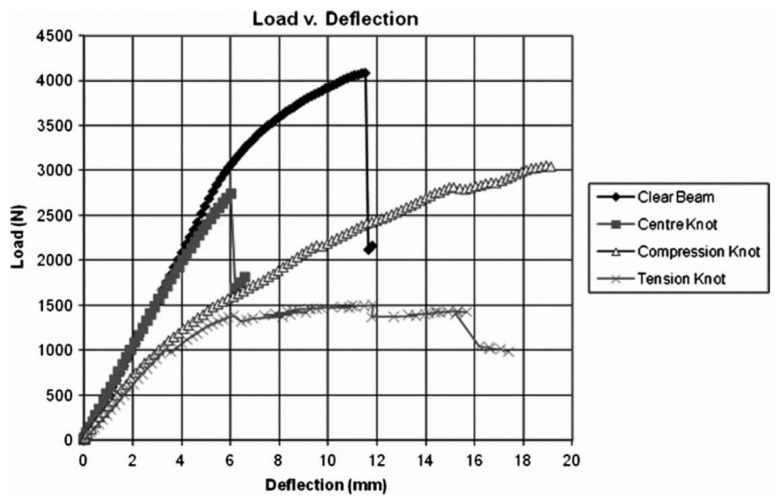
Plot of load version deflection for the beams shown in Figure 12. From [87].

**Figure 14 materials-15-05403-f014:**
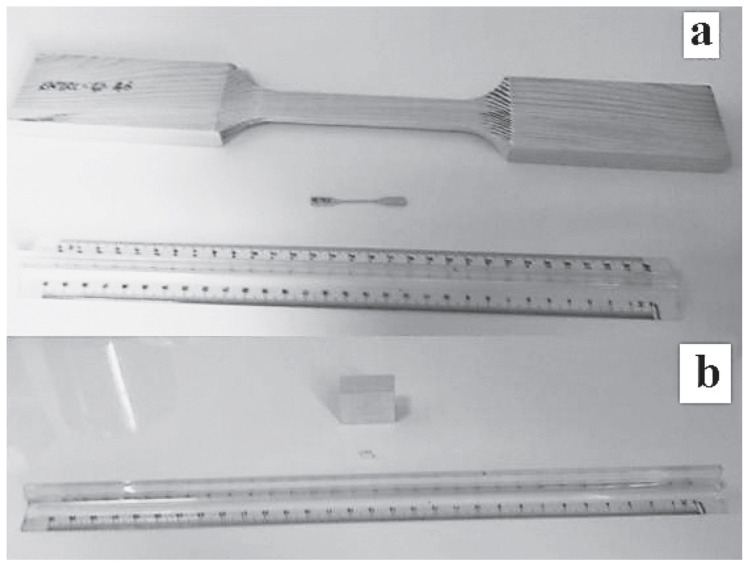
(**a**) Tensile and (**b**) compression specimens for investigating the size effect for wood. From [88].

**Figure 15 materials-15-05403-f015:**
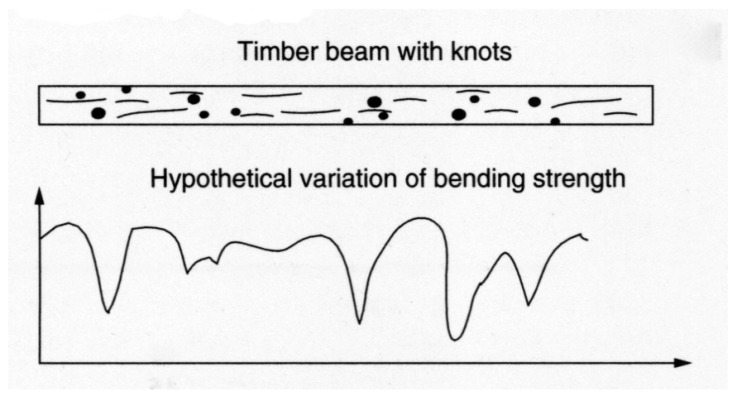
Schematic diagram showing how the strength of a wooden beam might vary along its length. From [100].

**Figure 16 materials-15-05403-f016:**
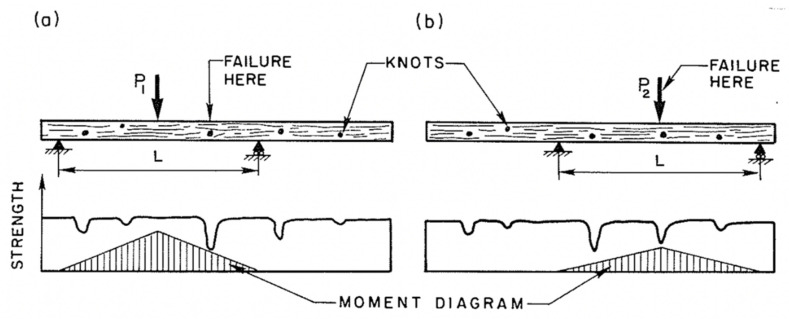
Schematic diagram showing how a wooden beam containing knots can fail at a different place from where it is loaded. From [102]. (**a**) Failure occurs away from peak load due to a weak knot close by. (**b**) Failure occurs at peak load as the peak load coincides with a knot, even though that knot is stronger than that shown in (**a**).

**Figure 17 materials-15-05403-f017:**
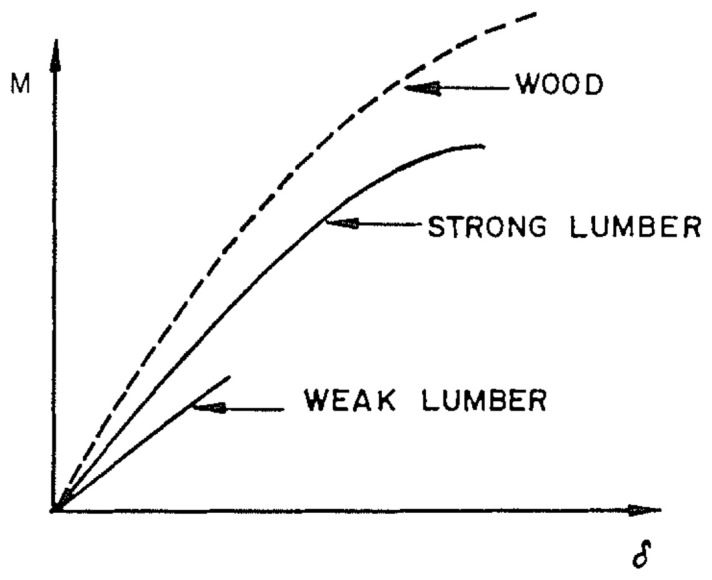
Schematic comparisons of load-deflection curves in bending for defect-free (clear) wood, ‘strong’ and ‘weak’ lumber. From [86].

**Figure 18 materials-15-05403-f018:**
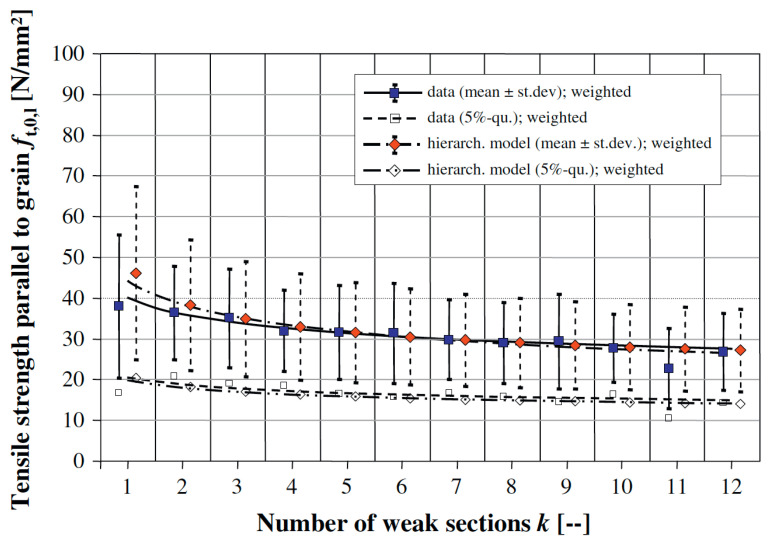
Plot of the tensile strength of a wooden beam against the number of weak sections that it contains. From [101].

**Figure 19 materials-15-05403-f019:**
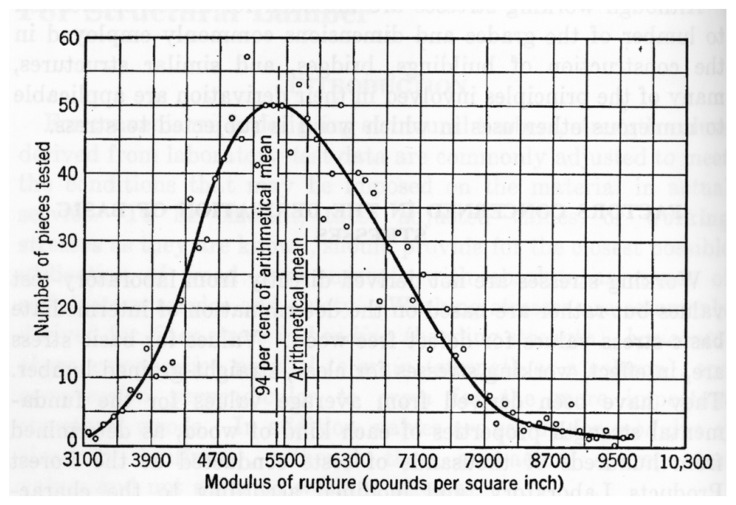
Variability in the modulus of rupture (MOR) of unseasoned clear-wood Sitka Spruce (*Picea sitchensis*). From [112]. White circles are data points.

**Figure 20 materials-15-05403-f020:**
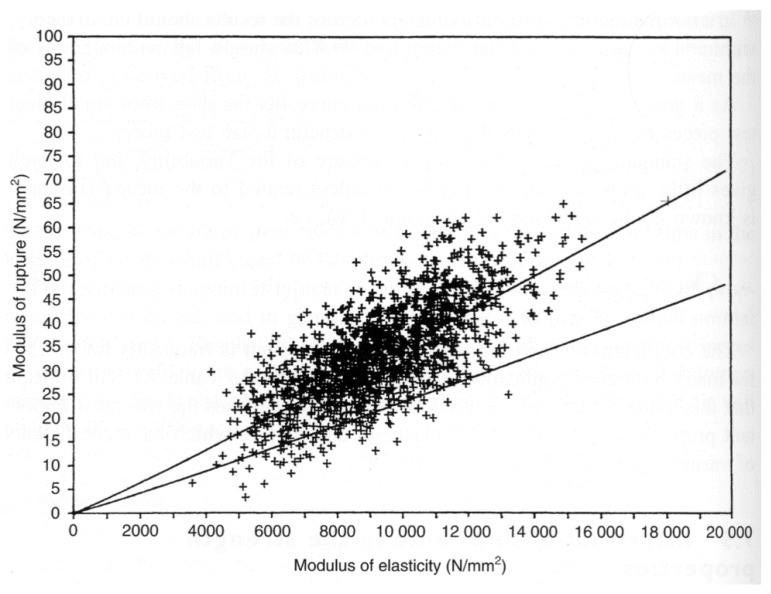
Data obtained from 1348 tests for the modulus of rupture (MOR) and the modulus of elasticity (MOE). The mean regression line and the 5th percentile exclusion line are plotted. The (over-precise) equation that Dinwoodie gave for the mean regression line was MOR = 0.002065(MOE)^1.0573^ with a correlation coefficient of 0.702. From [89]. ‘+’ signs are data points.

**Figure 21 materials-15-05403-f021:**
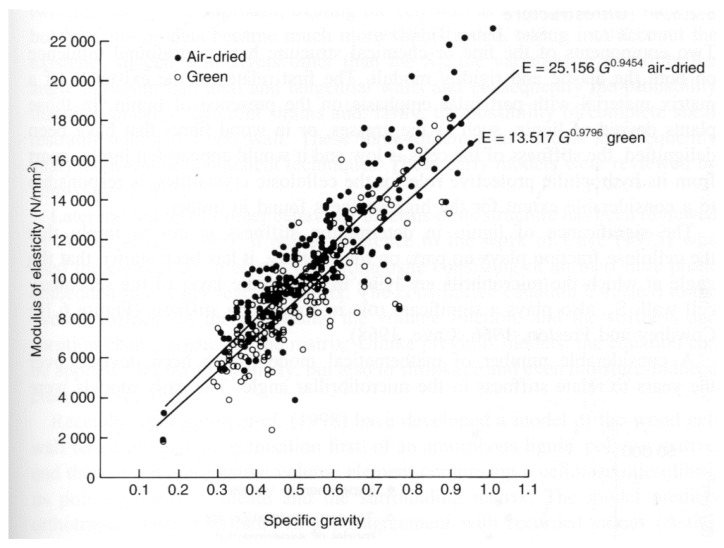
Plot of the longitudinal modulus of elasticity against specific gravity for more than 200 species of tree tested in green and dry states. From [92].

**Figure 22 materials-15-05403-f022:**
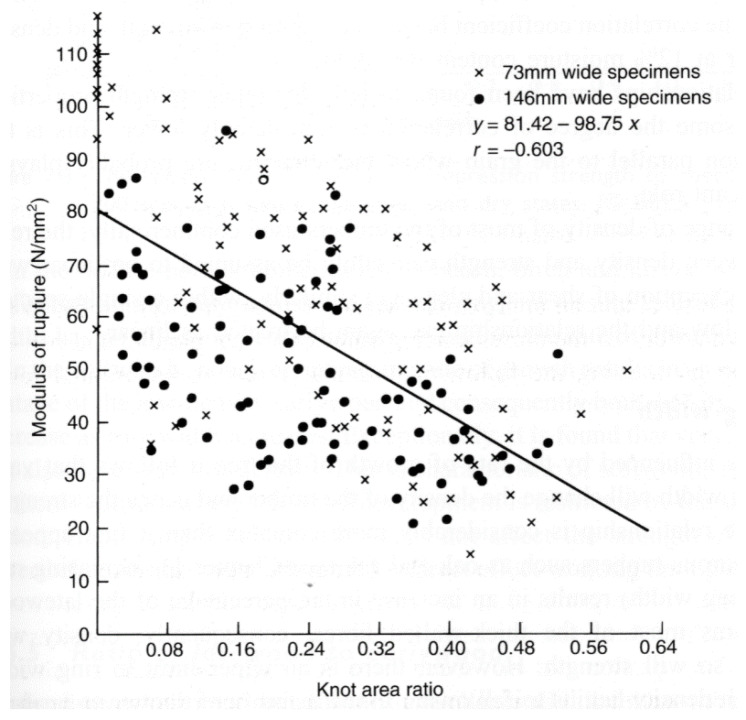
Effect of knot area ratio on the strength of Douglas Fir boards of two different widths. From [89]. The black circles and ‘x’ are data points.

**Figure 23 materials-15-05403-f023:**
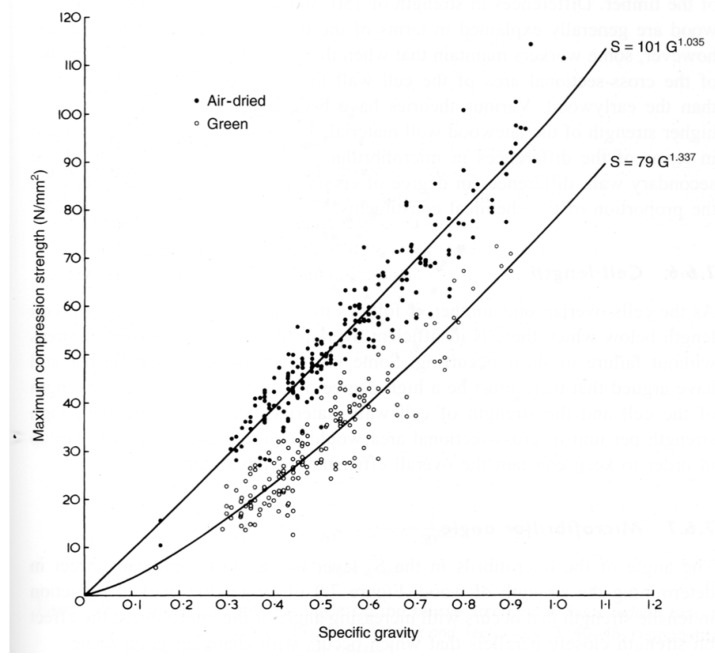
Plot of the maximum compression strength against specific gravity for wood taken from more than 200 species of tree, both green and air-dried. From [89].

**Figure 24 materials-15-05403-f024:**
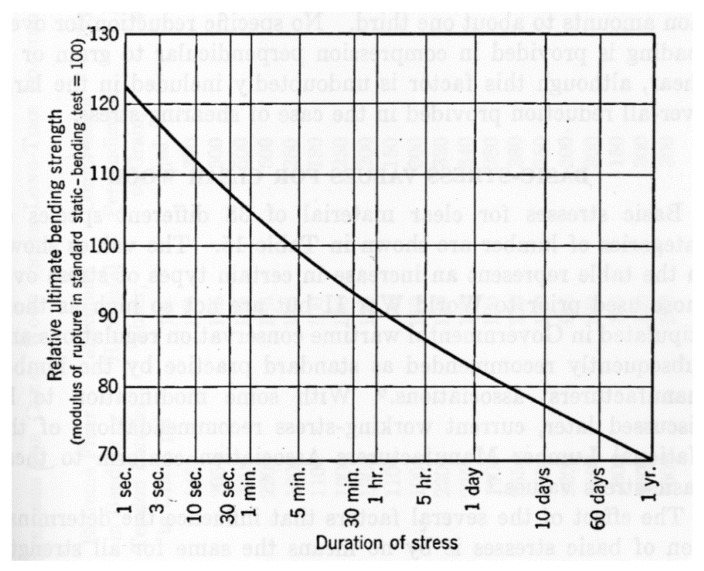
Variation in ultimate bending strength with duration of stress (time between initial application of load and failure) for unseasoned clear-wood Sitka Spruce (*Picea sitchensis*). From [112].

**Figure 25 materials-15-05403-f025:**
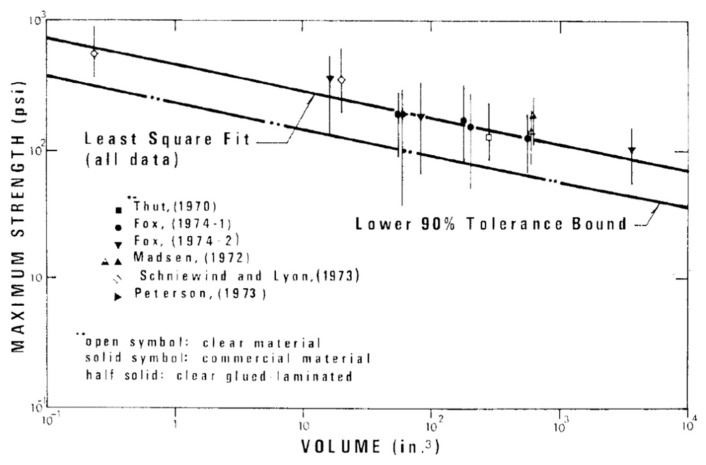
Relationship between strength and volume for uniformly loaded Douglas Fir blocks. From [113]. (References in the figure: ‘Thut (1970)’ [150], ‘Fox (1974)’ [151], ‘Madsen (1972)’ [152], ‘Schniewind and Lyon (1973)’ [153], ‘Peterson (1973)’ was a personal communication with Barrett.

**Figure 26 materials-15-05403-f026:**
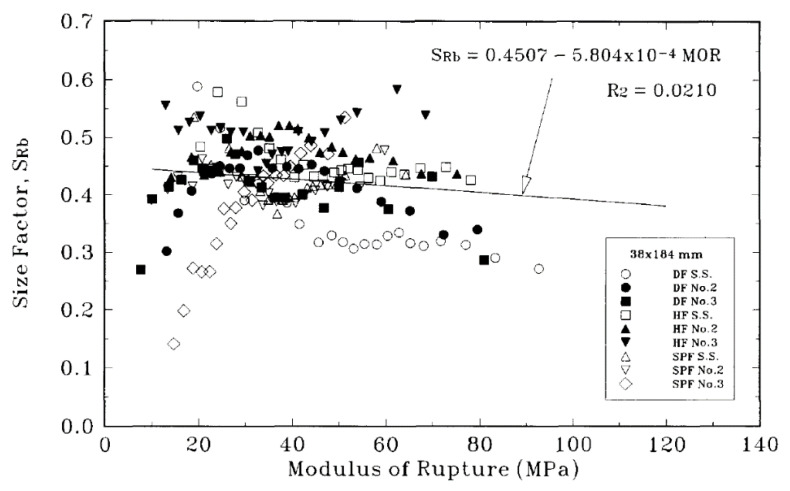
Plot showing the variation of bending size parameter S_Rb_ with modulus of rupture. From [154].

**Figure 27 materials-15-05403-f027:**
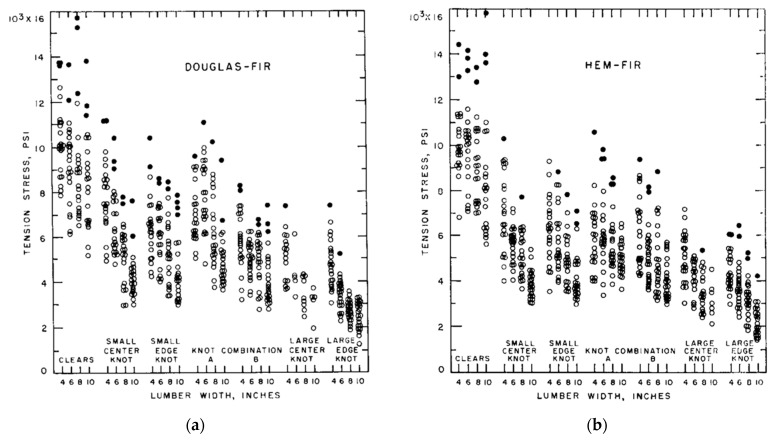
Plots of the measured tensile strengths for two-inch (5 cm) dimension lumber of four different nominal widths and seven different grades cut from two types of tree: (**a**) Douglas Fir and (**b**) Hem-Fir. From [155].

**Figure 28 materials-15-05403-f028:**
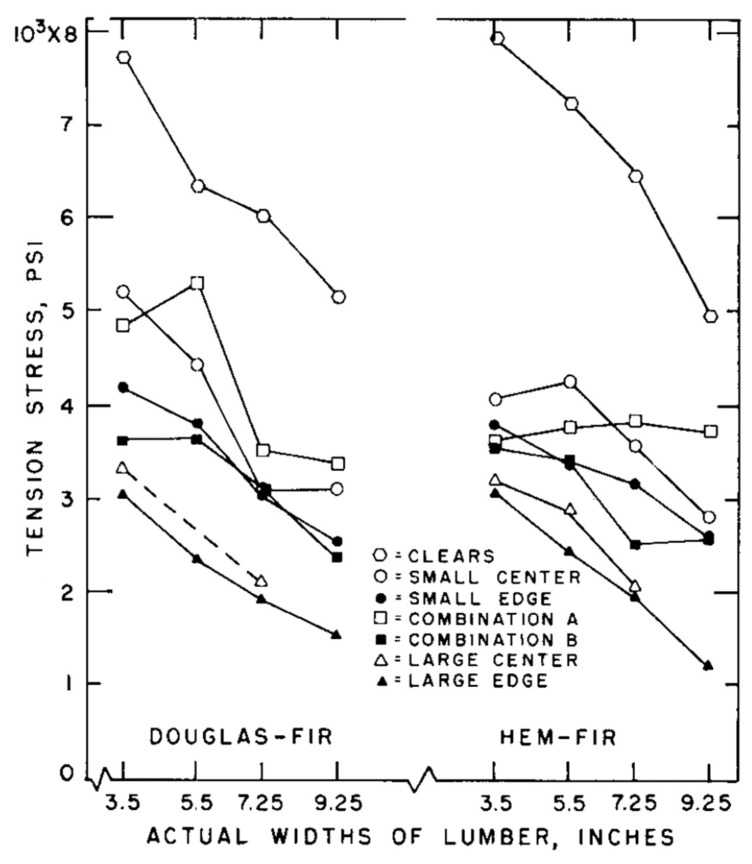
Plots of the tension stress values at the lower 5% exclusion limit (calculated from the data plotted in Figure 27) showing the size (width) effect more clearly. From [155].

**Figure 29 materials-15-05403-f029:**
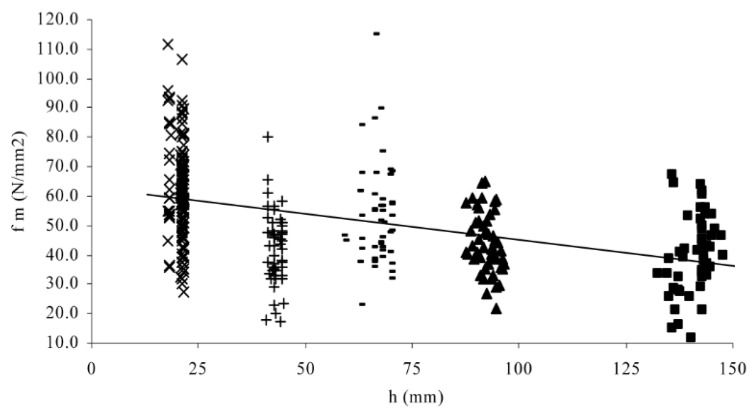
Plot of bending strength, f_m_, against specimen depth, h, of for 349 specimens. Note the high degree of scatter relative to variation in mean value between values of h. From [130]. The different symbols distinguish between specimens of different sizes.

**Figure 30 materials-15-05403-f030:**
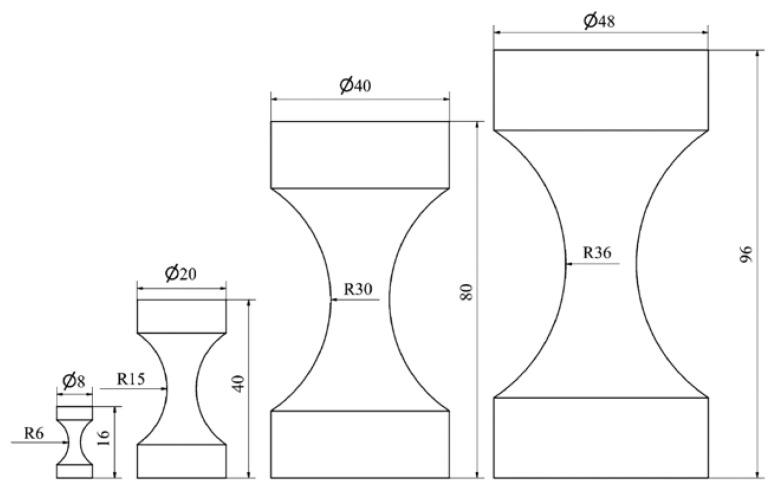
Cylindrically symmetric compression specimens used to investigate size effects in compression testing of Norway Spruce. Dimensions given in mm. From [125].

**Figure 31 materials-15-05403-f031:**
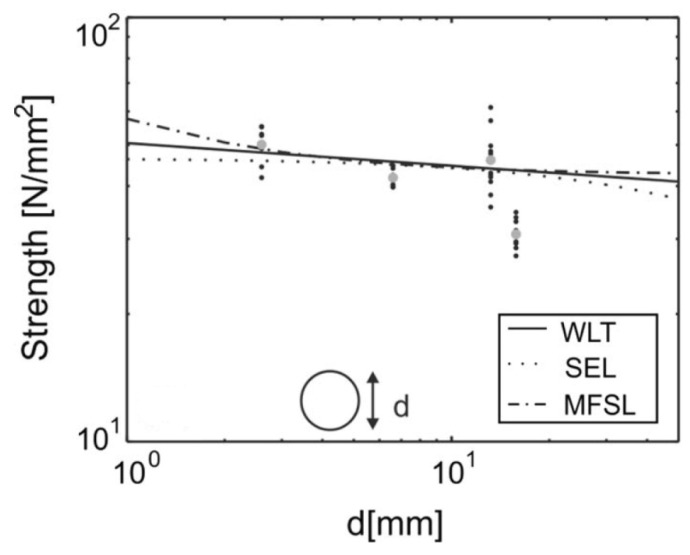
Plot of the effect of specimen size on compression strength. Comparison is made for three different theories. WLT means ‘weakest link theory’; SEL means ‘size effect law’, MFSL means ‘multifractal scaling law’. From [125].

**Figure 32 materials-15-05403-f032:**
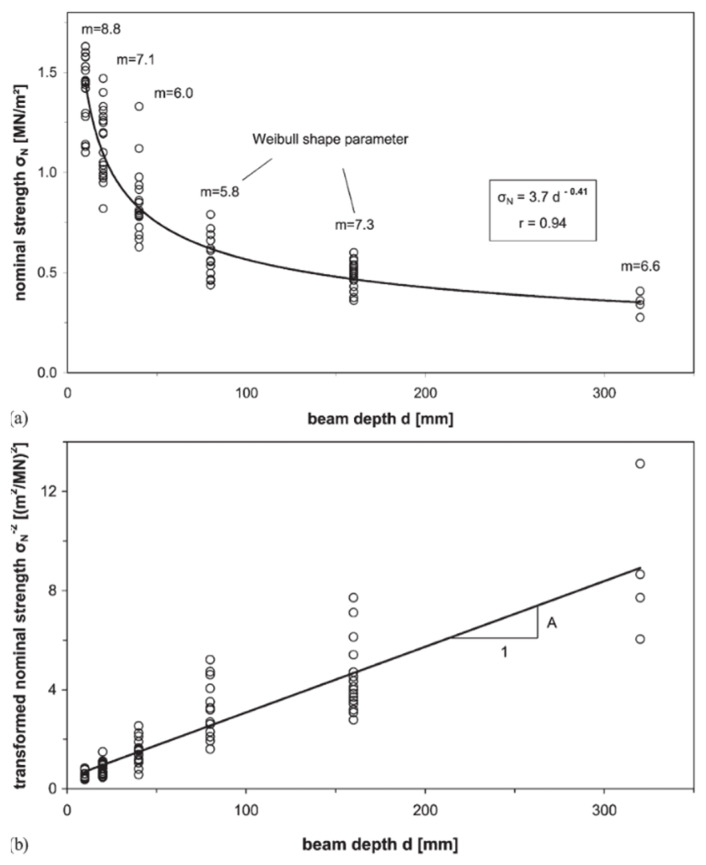
(**a**) Plot of bending strength, $\sigma_{N}$, of Spruce wood against beam depth, h. (**b**) Plot of $\sigma_{N}^{2}$ against h to test Bazant’s size effect law (see Equation (6) and [156]). From [162]. Circles are data points.

**Figure 33 materials-15-05403-f033:**
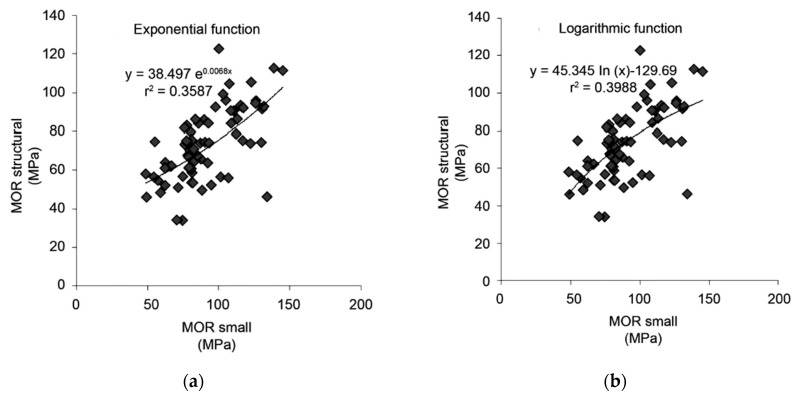
Plots of data obtained for modulus of rupture (MOR) for specimens of structural size against MOR for small specimens of mixed hardwoods. The lines in each plot represent various functions of structural size. (**a**) Exponential; (**b**) logarithmic; (**c**) power; (**d**) polynomial. From [160].

**Figure 34 materials-15-05403-f034:**
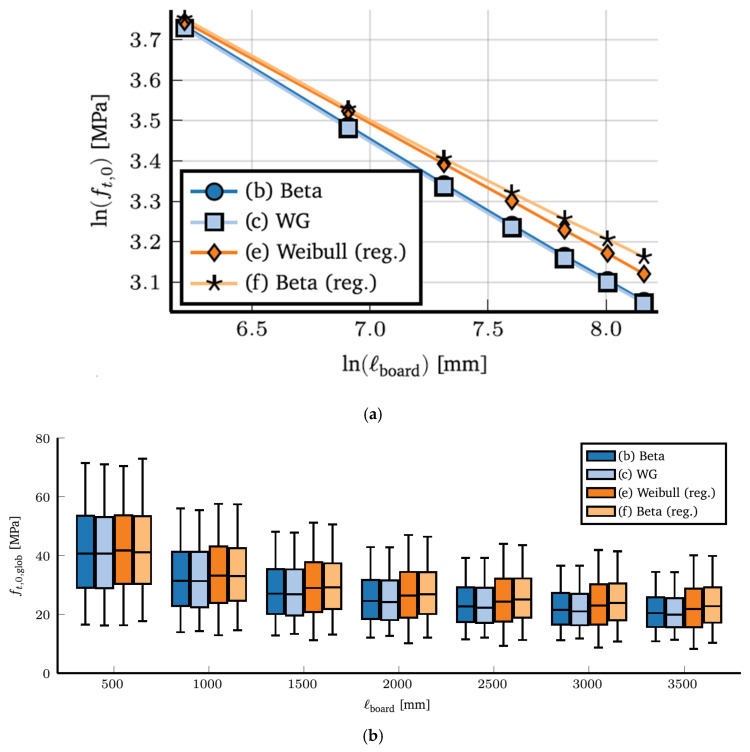
(**a**) Assessment of power law assumption for mean strength of European White Oak (*Quercus robur* and *Quercus petraea*) boards simulated by four different models. (**b**) Simulated length effect of tensile strengths obtained using four fitted models showing the variation for all grades studied. From [138].

**Figure 35 materials-15-05403-f035:**
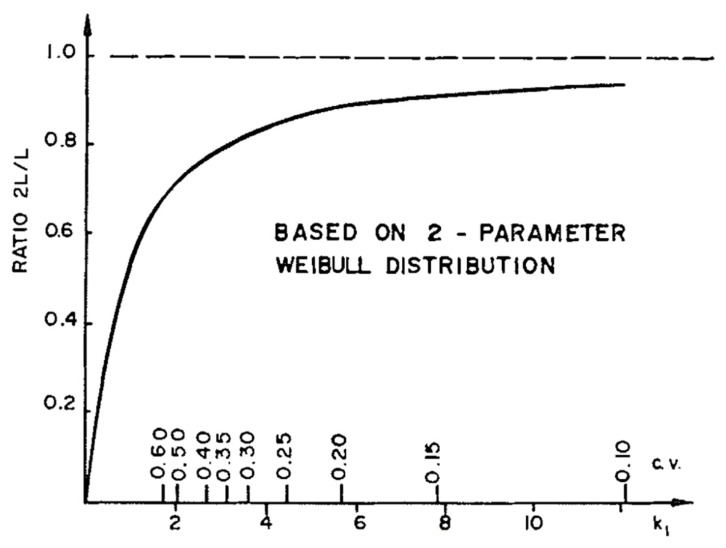
Plot of the equation discussed above. The axis labelled ‘c.v.’ plots the values of the coefficients of variation of strength within each board. This is related to $k_{1}$. From [86].

**Figure 36 materials-15-05403-f036:**
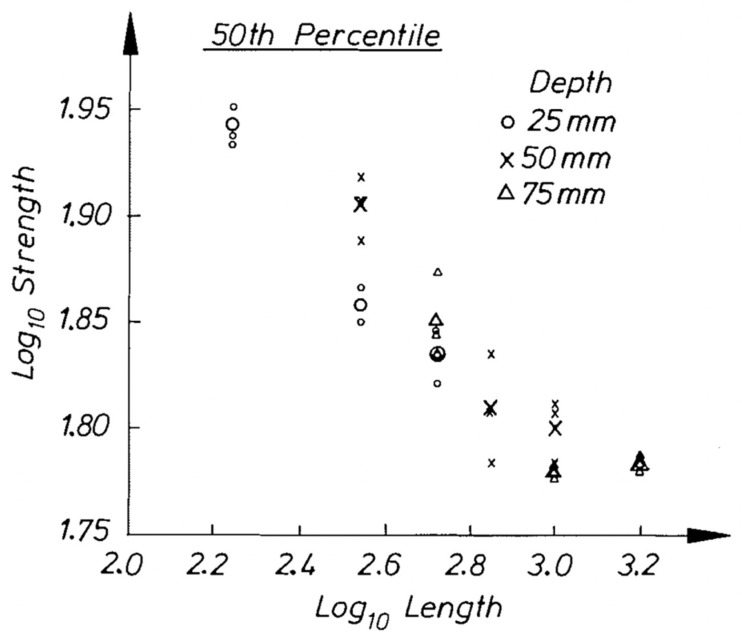
Plot of log(strength) versus log(length) for Canadian Spruce (*Picea glauca*). From [104].

**Figure 37 materials-15-05403-f037:**
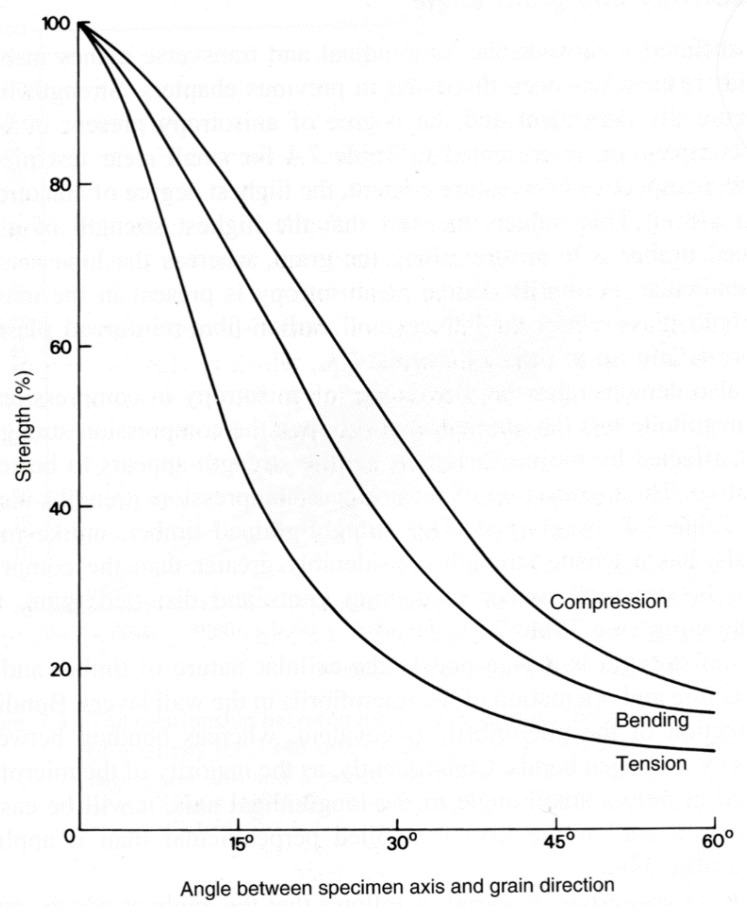
Effect of grain angle on the tensile, bending and compression strengths of timber. From [89].

**Figure 38 materials-15-05403-f038:**
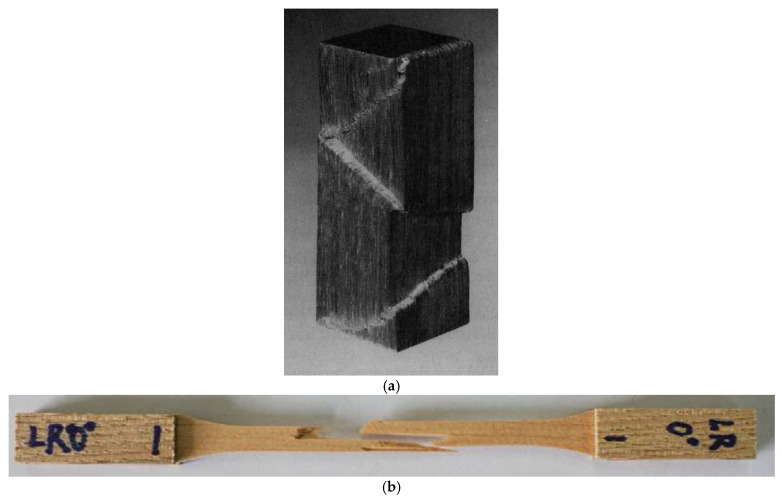
(**a**) Shear localization in a block of wood compressed longitudinally and parallel to the cellular structure. From [79]. (**b**) Example of shear fracture due to tensile loading of *Pinus koraiensis*. From [85]. (**c**) SEM and optical image of natural wood broken by bending. From [165].

**Figure 39 materials-15-05403-f039:**
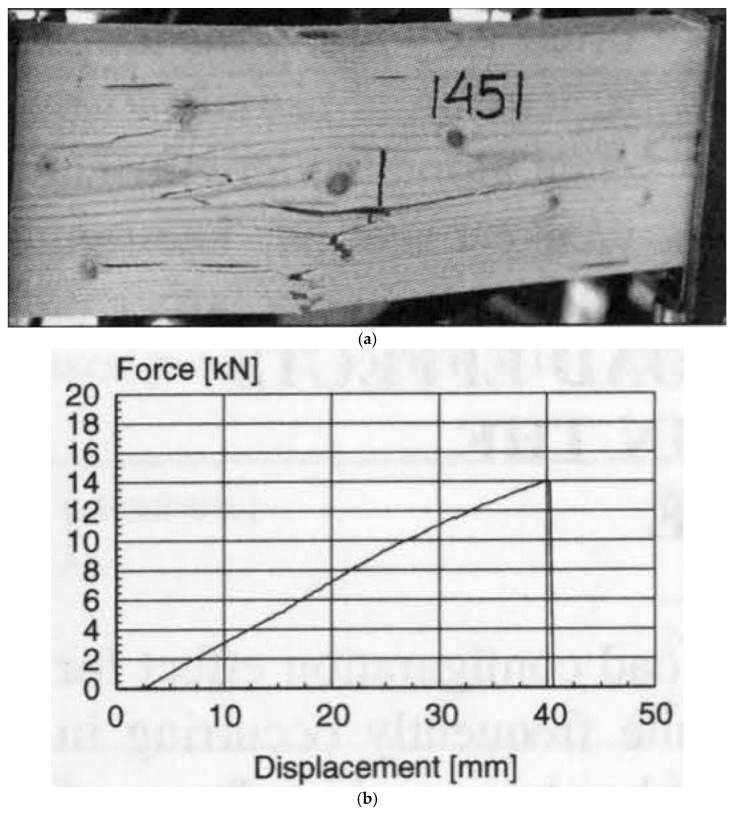
(**a**) Photograph of failed wood specimen that had been subjected to a bending load. (**b**) The load–displacement graph for the specimen shown in (**a**). The graph shows that no plastic deformation occurred before failure, i.e., the failure was brittle. From [100].

**Figure 40 materials-15-05403-f040:**
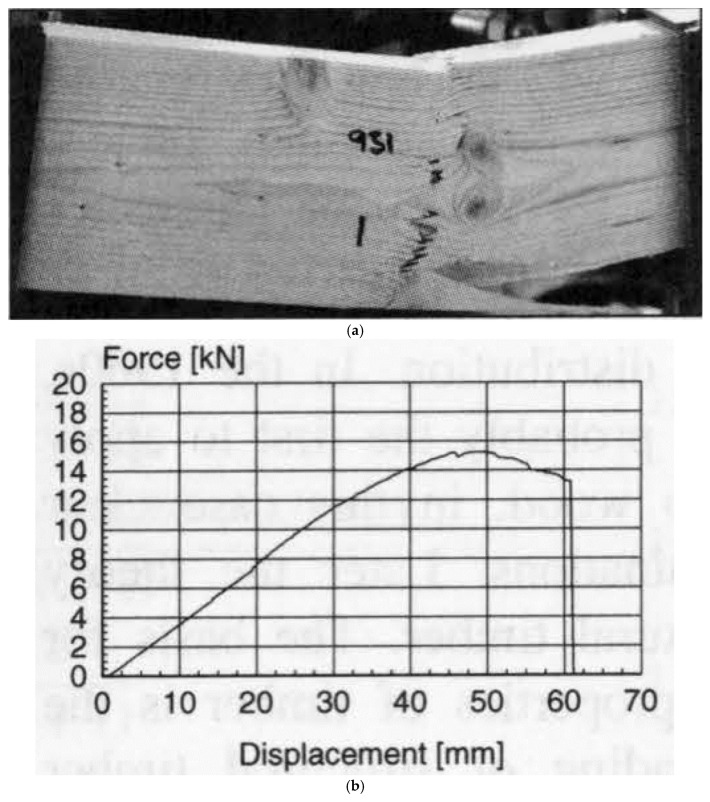
(**a**) Photograph of failed wood specimen that had been subjected to a bending load. (**b**) The load–displacement graph for the specimen shown in (**a**). The graph shows that plastic deformation occurred before failure, i.e., the failure was ductile. From [100].

**Figure 41 materials-15-05403-f041:**
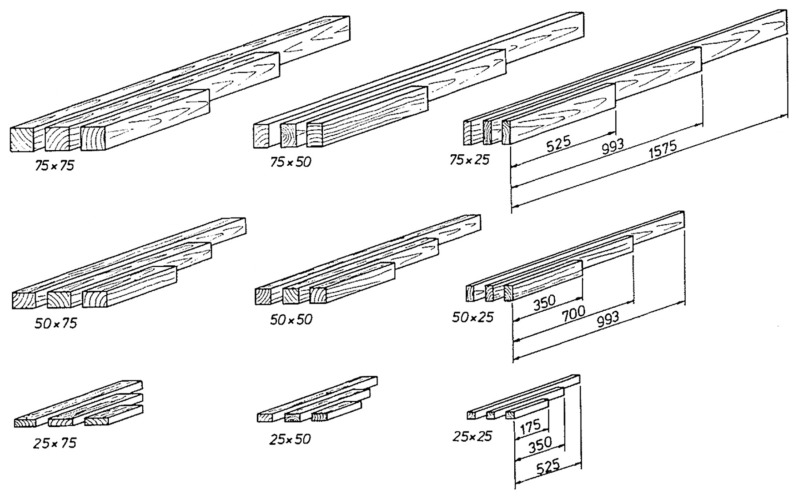
Total of 27 different specimen sizes for investigating length, depth, thickness size effects for spruce, pine, and fir. From [104].

**Figure 42 materials-15-05403-f042:**
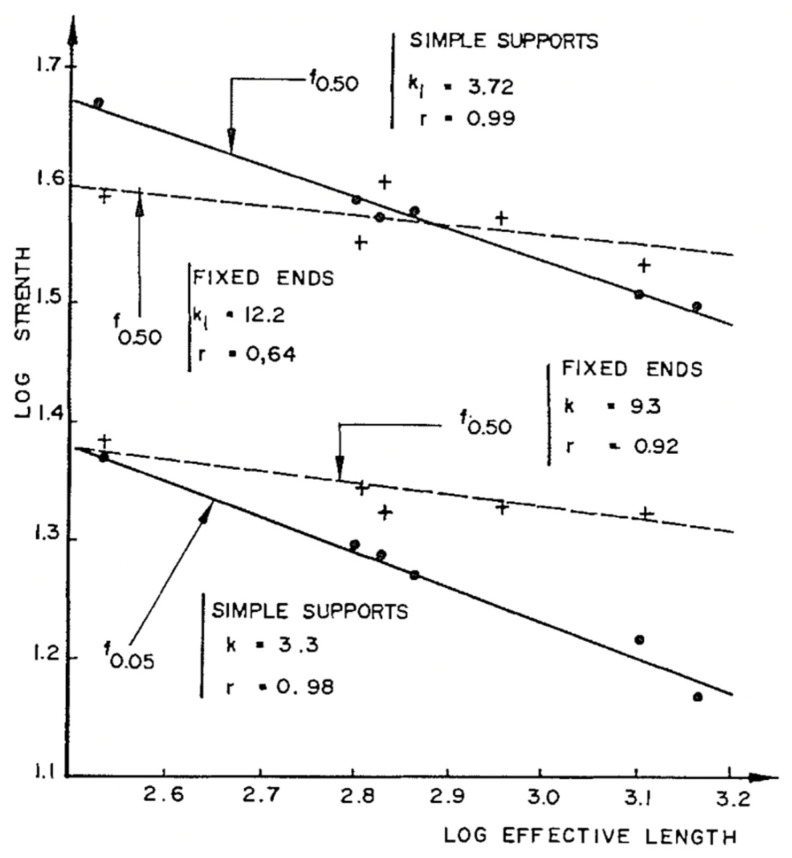
Plot showing the effect of length on the strength of wooden beams. From [86].

**Figure 43 materials-15-05403-f043:**
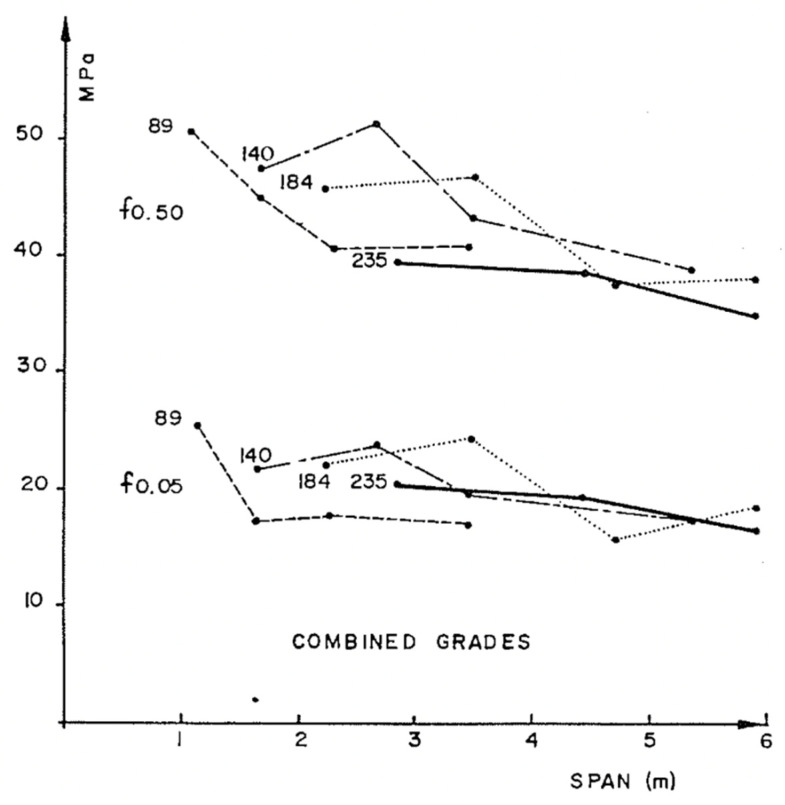
Fifth and fiftieth percentile strength data obtained from three-point loading of Hem-Fir wooden beams of the same span-to-depth ratio, but with different depths. These data were obtained in 1976. From [86].

**Figure 44 materials-15-05403-f044:**
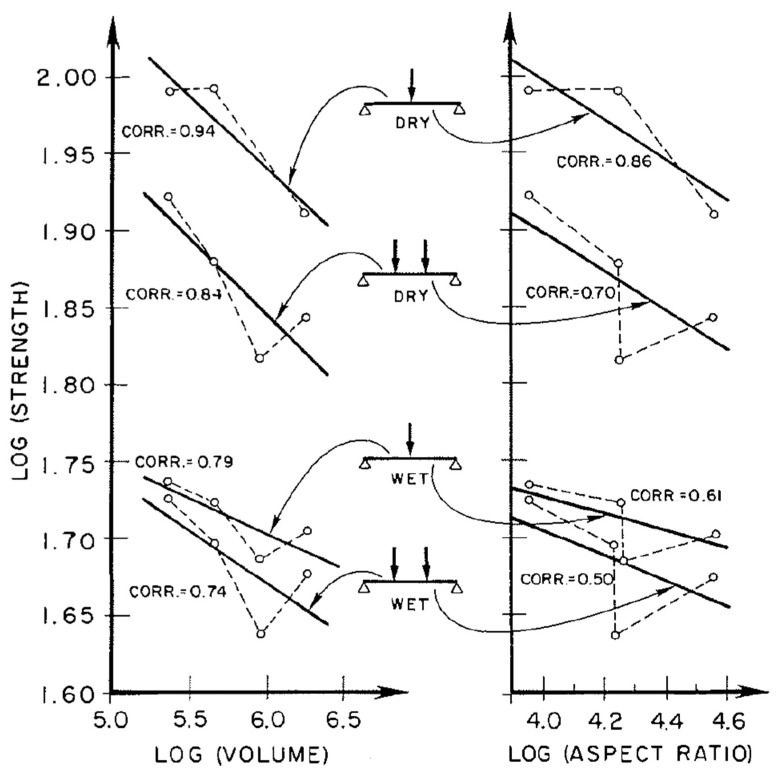
Plots of log(strength) vs. log(volume) and log(aspect ratio) for defect-free Douglas Fir. From [103].

**Figure 45 materials-15-05403-f045:**
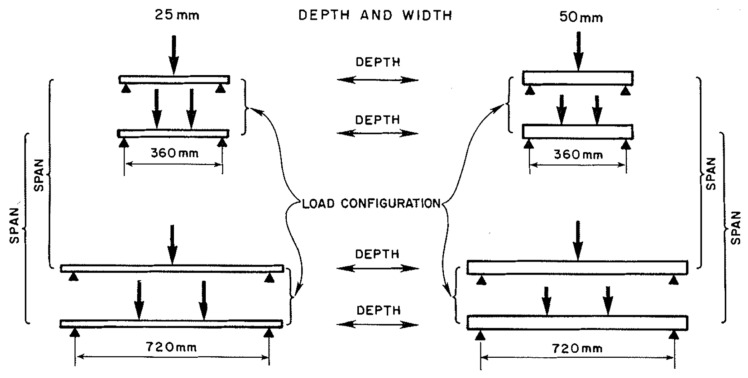
Various configurations of the load applied to the specimens shown in Figure 46. From [102].

**Figure 46 materials-15-05403-f046:**
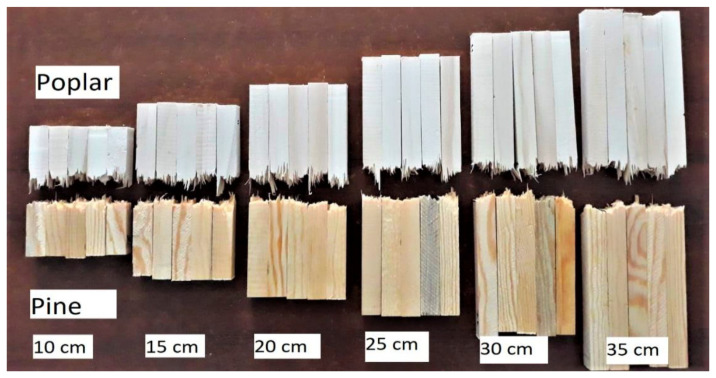
Photographs of groups of five test specimens of Poplar and Pine wood that had been subjected to an impact bending strength test. The labels give the span lengths of the original specimens in each group. From [166].

**Figure 47 materials-15-05403-f047:**
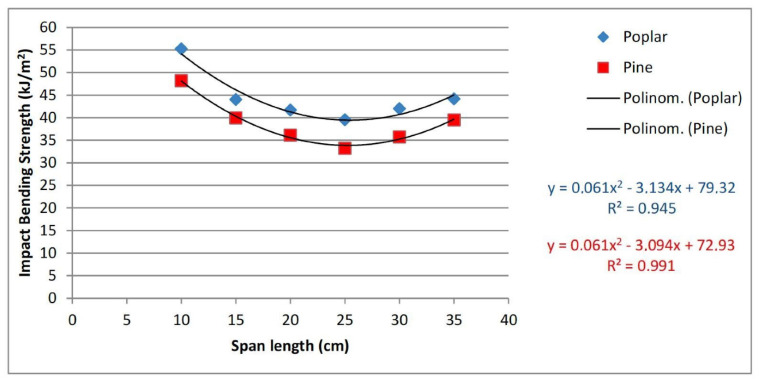
Plots of the impact bending strengths of Poplar and Pine specimens as a function of the span length of the original specimens. From [166].

**Figure 48 materials-15-05403-f048:**
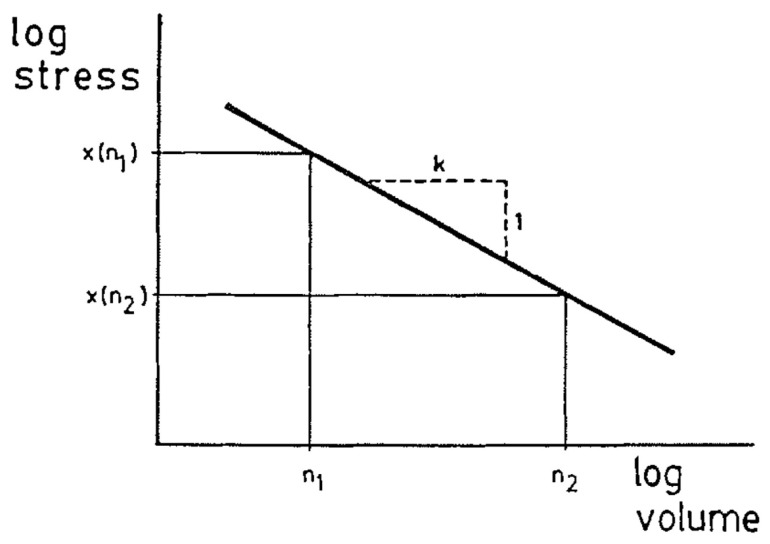
Schematic plot showing how the size effect parameter ‘g’ is calculated. From [86].

**Figure 49 materials-15-05403-f049:**
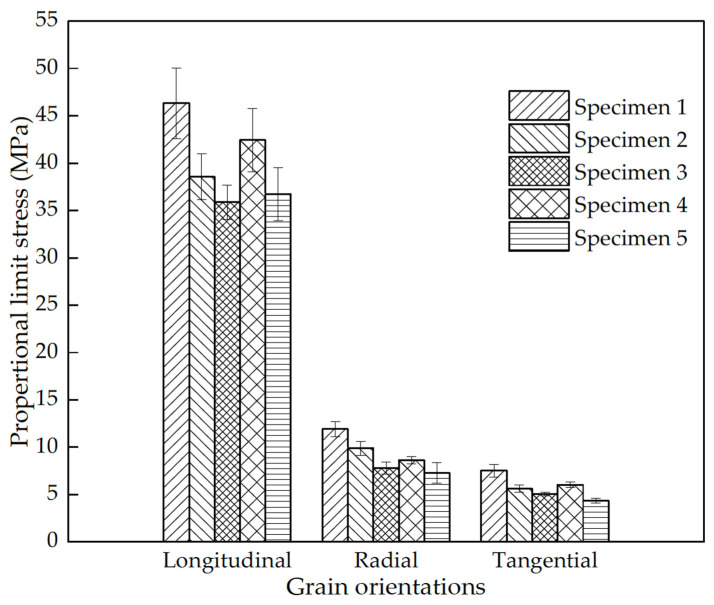
Proportional limit stresses for five different sizes of specimens of *Fagus sylvatica* cut in different orientations with respect to the trunk of the tree. From [134].

**Figure 50 materials-15-05403-f050:**
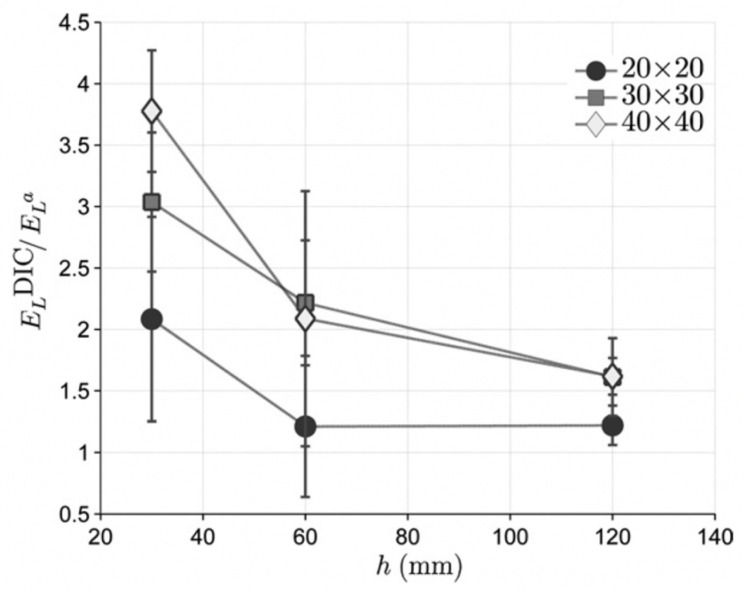
Plot of the ratio of elastic moduli measured two different ways for Maritime Pine specimens against specimen length for three different cross-sections. The two methods used were (i) optical (digital image correlation, DIC) and (ii) mechanical (a displacement transducer). From [167].

**Figure 51 materials-15-05403-f051:**
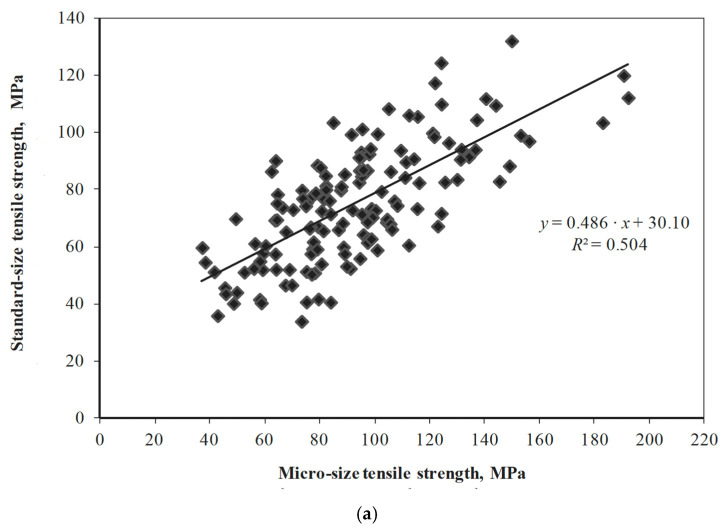
Comparison of the strengths of standard and micro-sized specimens of *Pinus sylvestris* in (**a**) tension (from [88]), (**b**) compression (from [88]) and (**c**) bending (from [168]).

**Figure 52 materials-15-05403-f052:**
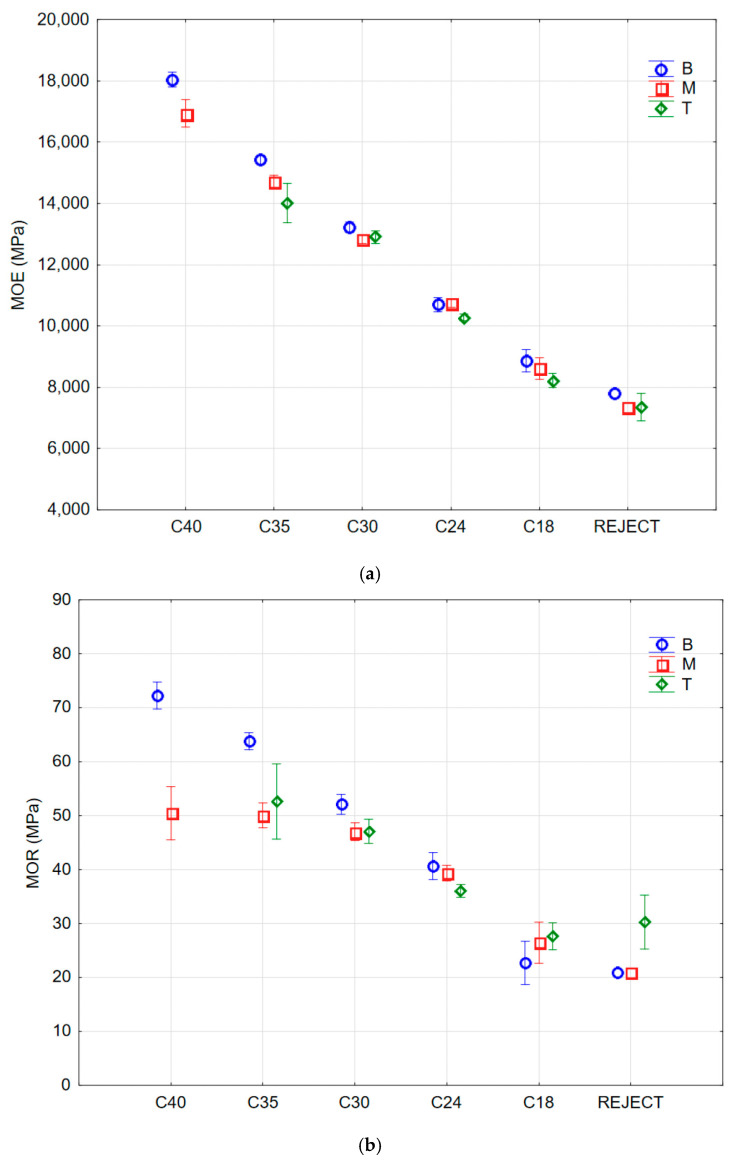
Plots of (**a**) Modulus of Elasticity (MOE) and (**b**) Modulus of Rupture (MOR) for logs of various grades (labelled C40 through to C18) of Scots Pine taken from various parts of the tree trunks (B, butt; M, middle; T, top). Data obtained using four-point bend experiments. From [169].

**Figure 53 materials-15-05403-f053:**
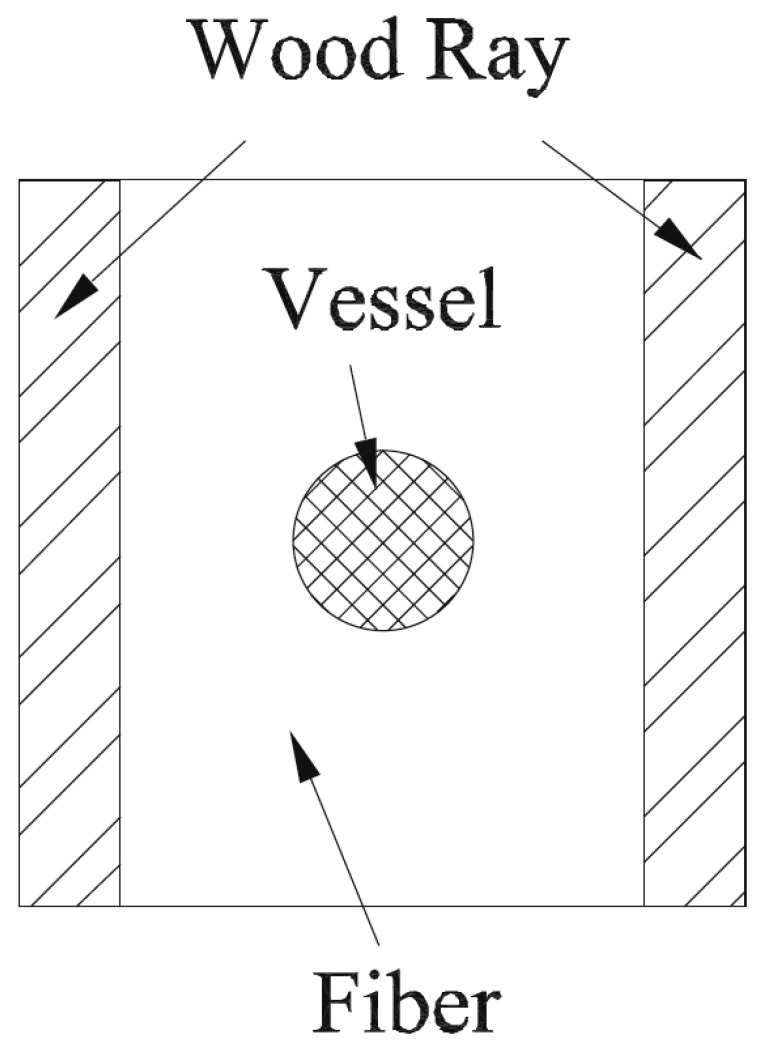
Simple representative volume element for a porous hardwood. From [13].

**Figure 54 materials-15-05403-f054:**
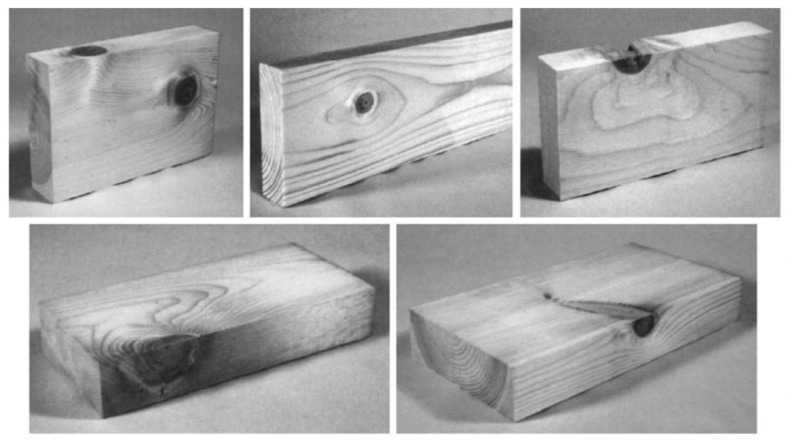
Main types of knots according to their position on the piece. Listing from left to right and top to bottom: edge and face knots, inner through knot, outer through knot, arris knot, splay knot. From [175].

**Figure 55 materials-15-05403-f055:**
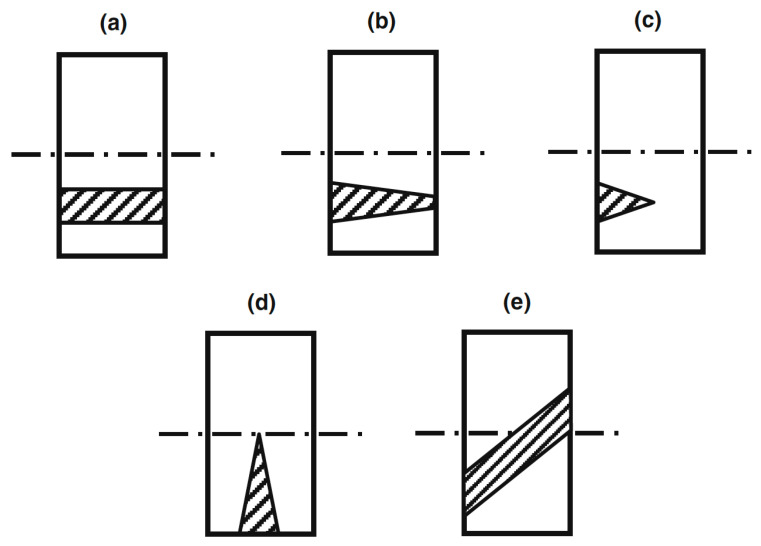
Schematic diagram showing the types of knots used in the parametric study: (**a**) cylindrical, (**b**) truncated conical, (**c**) shallow conical, (**d**) edge, and (**e**) inclined. From [178].

**Table 1 materials-15-05403-t001:** Early 19th century data (quoted to an absurd level of precision) on the failure loads of wrought iron, steel and timber pillars of various lengths and diameters. From [48].

Length.		Pillars with Both Ends Rounded.		Pillars with One End Flat, and the Other Rounded.		Pillars with Both Ends Flat.	
		Diameter.	Breaking Weight.	Diameter.	Breaking Weight.	Diameter.	Breaking Weight.
Wrought iron.	inches.	inch.	lbs.	inch.	lbs.	inch.	lbs.
	$90\frac{3}{4}$	1·017	1808	1·02	3355	1·02	5280
	$60\frac{1}{2}$	1·015	3938	1·03	8137	1·02	12,990
	$30\frac{1}{4}$	1·015	15,480	1·015	21,335	1·015	23,371
	$30\frac{1}{4}$	1·015	15,480	1·015	21,187 disc.	1·015	25,387 disc.
	$15\frac{1}{8}$	1·005	23,535	1·015	26,227	1·005	27,099
Steel.	29·95	·87	10,516	·87	20,135	·87	26,059
Timber.	$60\frac{1}{2}$	Side of square. 1·75	3197	Side of square. 1·75	6109	Side of square. 1·75	9625

1 inch is about 25 mm, 1 pound (lb.) is about 0.45 kg.

**Table 2 materials-15-05403-t002:** Key to the specimen dimensions for which data are plotted in Figure 49. From [134].

Specimen Label	Dimensions		
	Cross-Sectional Area/mm^2^		
	Width/mm	Length/mm	Height/mm
1	10	10	10
2	10	10	20
3	10	10	30
4	20	20	30
5	30	30	30

**Table 3 materials-15-05403-t003:** Average values and standard deviations of the longitudinal modulus of elasticity for various lengths and cross-sections of specimens of *Pinus pinaster.* From [167].

Cross-Section/mm^2^	Height/mm		
	30	60	120
20 × 20	15.7 ± 2.7 GPa	15.9 ± 3.1 GPa	14.5 ± 2.0 GPa
30 × 30	16.9 ± 2.9 GPa	15.1 ± 3.0 GPa	15.1 ± 2.9 GPa
40 × 40	18.1 ± 1.7 GPa	16.1 ± 2.7 GPa	15.8 ± 2.3 GPa

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
