# Peer review of "Is Wood a Material? Taking the Size Effect Seriously"

_materials, 2022, doi:10.3390/ma15155403_

Round 1
Reviewer 1 Report
This review manuscript highlights the differences between homogeneous materials
and structures in the specific case of wood. Given the importance of structured,
inhomogeneous materials for today's high performance engineering requirements,
this review provides an excellent introduction to this topic. The difficulties in
determining model parameters such as stiffness and strength are discussed, given
natural variability. Particular emphasis is put on size effects on failure, with
easy to understand shortened recapitulation of the most important failure
theories due to Weibull and Bazant. The manuscript is very well written.
This work was a joy to read. As a researcher working in testing of both
materials and structures, I can attest to the scientific correctness of what has
been written. However, I am no expert in the mechanical behaviour of wood, and I
have learned many things from this review. I recommend publication, as I see the
benefit of this work for other researchers. No corrections are required.
Author Response
See attached pdf.

Reviewer 2 Report
Comments and Suggestions for Authors
Based on previous studies, this manuscript summarized that wood size has a significant effect on mechanical properties. The title is novel, but it needs further revision. In addition, there are still many problems in this work, and further improvement is needed to meet the requirements of publication.
1. The title should be changed because it is not too much related to the contents of the review manuscript. It is suggested to be more focused on the structure and mechanical propertied of wood. In addition, is it correct to only study the mechanical properties of wood and judge whether wood is a material?
2. The two authors are from the same lab. It is suggested to only provide one affiliation on title page.
3. More introduction on the microstructures of wood is suggested with more recent articles: MOFs meet wood: Reusable magnetic hydrophilic composites toward efficient water treatment with super-high dye adsorption capacity at high dye concentration; Design of wood-derived anisotropic structural carbon electrode for high-performance supercapacitor.
4. The authors must have their work reviewed by a proper translation/reviewing service before submission; only then can a proper review be performed. Most sentences contain grammatical and/or spelling mistakes or are not complete sentences.
5. In this sentence: “In this review we will take the term ‘material’ to mean a substance that has mechanical properties that are independent of the size of the object made from it [9].” The title and the word "material" in the text should in the same format, with “ ”.
6. The writing format of (a), (b).... should be uniform. For example, Figure 5 is different from Figure 6.
7. The quality and clarity of all the figures are poor. All the figures should be modified to provide a better resolution, and the word size of the picture should be adjusted.
8. In the part of: “3 Problems with the Application of Elasticity Theory to Wood”. Because of the presence of knots, the mechanical properties of wood are different. However, the relationship between knot and wood size is not described in detail in the manuscript. And, the influence of wood size on wood mechanical properties is only emphasized in the entire manuscript.
9. Two important articles regarding the mechanical behavior should be considered in the manuscript: Mechanical behaviour of wood compressed in radial direction-part I. New method of determining the yield stress of wood on the stress-strain curve; Mechanical behaviour of wood compressed in radial direction: Part II. Influence of temperature and moisture content.
10. The effect of size on the mechanical properties of wood has been pointed out several times. However, too much repetition of the same content should be avoided.
11. In the part of “Conclusions and Matters for Further Study”, some research significance of the review should be included. In addition, it should provide the reference value in the specific research field and the prospect of the future related work.
12. There are too many too old references in the manuscript. Some more recent, relevant and important articles regarding wood structures and mechanical performances should be added into the manuscript.
13. There are many formatting errors in the references, which should be further modified to meet the requirements of the journal.
Author Response
Based on previous studies, this manuscript summarized that wood size has a significant effect on mechanical properties. The title is novel, but it needs further revision. In addition, there are still many problems in this work, and further improvement is needed to meet the requirements of publication.
First, although we disagree with many of the points raised by this referee, we would like to thank them for the care with which they read and assessed our paper.
- The title should be changed because it is not too much related to the contents of the review manuscript. It is suggested to be more focused on the structure and mechanical propertied of wood. In addition, is it correct to only study the mechanical properties of wood and judge whether wood is a material?
Response: We disagree. The aim of the choice of title is to intrigue and to provoke our readers. Only mechanical properties were considered as it is for these that the size effect is most pronounced and most studied. We wanted to alert people to the serious implications of not taking the size effect in wood seriously. This can result in the invalid extrapolation of laboratory measurements of mechanical properties of small specimens of wood to the large structural components used in buildings.
- The two authors are from the same lab. It is suggested to only provide one affiliation on title page.
Response: the reason two separate affiliations were given is that our emails are different.
- More introduction on the microstructures of wood is suggested with more recent articles: MOFs meet wood: Reusable magnetic hydrophilic composites toward efficient water treatment with super-high dye adsorption capacity at high dye concentration; Design of wood-derived anisotropic structural carbon electrode for high-performance supercapacitor.
Response: only the properties of wood relevant to its use in construction have been considered.
- The authors must have their work reviewed by a proper translation/reviewing service before submission; only then can a proper review be performed. Most sentences contain grammatical and/or spelling mistakes or are not complete sentences.
Response: We are surprised that the referee made this recommendation as higher up the review form, the referee put an ‘x’ in the box indicating that ‘I don’t feel qualified to judge about the English language and style’. Also the authors of this paper are both native speakers and writers of English.
- In this sentence: “In this review we will take the term ‘material’ to mean a substance that has mechanical properties that are independent of the size of the object made from it [9].” The title and the word "material" in the text should in the same format, with “ ”.
Response: ‘scare quotes’ should be used sparingly. We put the word ‘material’ in quotes in the text at that point to indicate that it is this word or concept we are talking about. Using quote marks in the title would, we feel, detract from our intent to intrigue and provoke our readers (see above).
- The writing format of (a), (b).... should be uniform. For example, Figure 5 is different from Figure 6.
Response: We agree that the labelling of Figure 5 looks messy. But cutting the figure in two and relabelling would risk breaking the link between the left- and right-hand graphs.
- The quality and clarity of all the figures are poor. All the figures should be modified to provide a better resolution, and the word size of the picture should be adjusted.
Response: The quality of the figures cannot be improved as they were either obtained by taking screen-shots at maximum resolution from online pdfs or scanned at high resolution from books.
- In the part of: “3 Problems with the Application of Elasticity Theory to Wood”. Because of the presence of knots, the mechanical properties of wood are different. However, the relationship between knot and wood size is not described in detail in the manuscript. And, the influence of wood size on wood mechanical properties is only emphasized in the entire manuscript.
Response: Knots are treated in the wood literature as large and visible defects (see, for example, Figures 15 & 16). Thus knots are the main reason historically why the Weibull distribution was chosen as describing the size effect in wooden beams. As mechanical data for wood has a great deal of experimental scatter, we don’t believe a more sophisticated analysis can be justified.
- Two important articles regarding the mechanical behavior should be considered in the manuscript: Mechanical behaviour of wood compressed in radial direction-part I. New method of determining the yield stress of wood on the stress-strain curve; Mechanical behaviour of wood compressed in radial direction: Part II. Influence of temperature and moisture content.
Response: Very many thanks for bringing these interesting papers (and the journal they were published in) to our attention. However, we do not feel they are relevant to the main point of the paper, which is the implication of the size effect on the use of results obtained from the mechanical testing of wood. These two papers are certainly relevant to the way wood may be modified in future to make it more uniform and hence mitigate the effect of specimen size on mechanical properties. So if we were to cite them, it would be in the ‘conclusions and suggestions for further studies’ section.
- The effect of size on the mechanical properties of wood has been pointed out several times. However, too much repetition of the same content should be avoided.
Response: The effect of size on the mechanical properties of wood was the main point of the paper. That is why we kept returning to this theme throughout the paper. Sorry if we overdid it.
- In the part of “Conclusions and Matters for Further Study”, some research significance of the review should be included. In addition, it should provide the reference value in the specific research field and the prospect of the future related work.
Response: In the ‘Conclusions’ section we have stated we have demonstrated what we set out to show, namely that wood should not be considered to be a material when it comes to mechanical testing. About the matter of ‘Further study’ we have made a suggestion about how this idea should affect the mechanical testing of wood in the future.
- There are too many too old references in the manuscript. Some more recent, relevant and important articles regarding wood structures and mechanical performances should be added into the manuscript.
Response: We disagree. We are up to date with the wood literature, but the aim of this paper was not to write a state-of-the-art review of the present knowledge of wood’s mechanical properties but to alert people to the implications of the intrinsic size effect for wood. We have now performed an analysis of the dates of publication of the citations in this paper: 72 were published between 2010 and 2022, 21 were published between 2000 and 2009, 9 were published between 1990 and 1999, 73 were published before 1990. So, if anything, the citations are bimodal between the historic literature (which we feel it is useful to bring to people’s attention) and the most modern publications.
- There are many formatting errors in the references, which should be further modified to meet the requirements of the journal.
Response: the references were formatted using the MDPI style guide. If we have done this incorrectly, we trust that the editorial staff of the journal will bring the citations into line with their policy (or ask us to reformat).

Reviewer 3 Report
This paper deals with wood's elastic and inelastic properties and models according to which they were/are determined. Numerous findings of other authors have been used and properly cited in this manuscript. Although it is quite extensive and, in that case, more appropriate to be published as a book chapter, given its quality, the manuscript can/should be published in a form of a review article. The topic is quite interesting and the manuscript is prepared well, although some parts of it could represent a challenge for readers not so familiar with the viscoelasticity and wood's mechanical properties. Also, some of the claims stated in this manuscript could surprise many other researchers and that is quite interesting, so hopefully this work will have its continuation.
Author Response
This paper deals with wood's elastic and inelastic properties and models according to which they were/are determined. Numerous findings of other authors have been used and properly cited in this manuscript. Although it is quite extensive and, in that case, more appropriate to be published as a book chapter, given its quality, the manuscript can/should be published in a form of a review article. The topic is quite interesting and the manuscript is prepared well, although some parts of it could represent a challenge for readers not so familiar with the viscoelasticity and wood's mechanical properties. Also, some of the claims stated in this manuscript could surprise many other researchers and that is quite interesting, so hopefully this work will have its continuation.
Response: Thank you for your kind words. The aim of writing this article was to intrigue and provoke. If we cause surprise to some of our readers, we will have achieved one of our objectives.

Round 2
Reviewer 2 Report
Authors did not response properly to most of issues. The revision did not any improvement to the previous version, including the content, the figures, the proposed concept, etc.
Author Response
The reason we did not modify the paper in line with the referee's comments is that we disagreed with them. For example, since the figures were obtained either from pdf's published online or scanned from books, their quality cannot be improved. Also the referee was critical of the standard of English despite declaring themselves not qualified to judge the language or style.